# Improving Discrete Diffusion with Schedule-Conditioning

## Abstract

Discrete diffusion models, like continuous diffusion models, generate high-quality sequence data by gradually undoing noise applied to datapoints via a Markov process. Gradual generation in theory comes with many conceptual benefits; for example, inductive biases can be incorporated into the noising Markov process. In practice however, the best performing discrete diffusion model is consistently masking, which does not denoise gradually. Here we explain the performance of masking diffusion by noting that it makes use of a fundamental difference between continuous and discrete Markov processes: discrete Markov processes evolve by discontinuous jumps at a fixed rate and, unlike other discrete diffusion models, masking diffusion *builds in the known distribution of jump times* and only learns where to jump to. We show that we can similarly bake in the known distribution of jump times into *any* discrete diffusion model; despite their simplicity, our new models – schedule-conditioned diffusion (SCUD) – generalize classical discrete diffusion and masking diffusion. By applying SCUD to models with noising processes that incorporate inductive biases on images, text, and protein data, we build diffusion models that outperform masking.

## 1 Introduction

Discrete diffusion models are state of the art models for conditional generation of discrete sequences. In biological sequence design, for example, they allow one to generate sequence flexibly conditioned on protein structure (Luo et al., 2022), DNA function (Sarkar et al., 2024), protein family (Alamdari et al., 2023), and other properties (Gruver et al., 2023; Nisonoff et al., 2024). They are also nearing state-of-the-art generation on language data (Sahoo et al., 2024). To define a diffusion model, one proposes a "forward" process by which data is gradually transformed token-by-token into noise and then learns a "backward" transformation that turns noise into data by optimizing an ELBO. In principle, the quality of the learned model should benefit from a forward process that captures structure in the data distribution. For example, works have suggested forward processes that are more likely to transform tokens into similar tokens – therefore the noising process is more "gradual" (Austin et al., 2021; Alamdari et al., 2023) ; as well as "state-dependent" processes that transform certain tokens more quickly than others (Shi et al., 2024). Surprisingly, these methods are all outperformed by "masking diffusion" which has the simplest possible forward process – one transforms each token into a masking token at a uniform rate (Austin et al., 2021; Alamdari et al., 2023; Shi et al., 2024).

Here we propose that this is because masking diffusion benefits from a parameterization that forces the distribution of corruption / transition events, the "transition schedule", in the backward process to match the distribution in the forward process. We use this insight to build models that unlock the benefits of structured and state-dependent processes in practice. First in Sec. 3 we provide a new decomposition of the ELBO that includes a term describing the mismatch in the distribution of the schedules of the forward and backward processes. Then in Sec. 4 we describe how to efficiently train models that build in the transition schedule (Fig. 1) to set this term to 0. We call our models schedule conditioned diffusion (SCUD). In Sec. 5 we show that when SCUD is applied to discrete diffusion with a uniform forward process, the result is masking diffusion, explaining its superior performance. Finally in Sec. 7 we unlock the potential of structured and state-dependent discrete diffusion by building SCUD versions of these methods and see that they finally beat masking diffusion (Fig 2). We release our code at https://anonymous.4open.science/r/SCUD-3844/.

## 2 Background

Our goal is to model data from a distribution $p(x_0)$ where $x_0$ is a sequence of discrete elements that belong to a set of size $B$. First we consider the one-dimension case and consider sequences later.

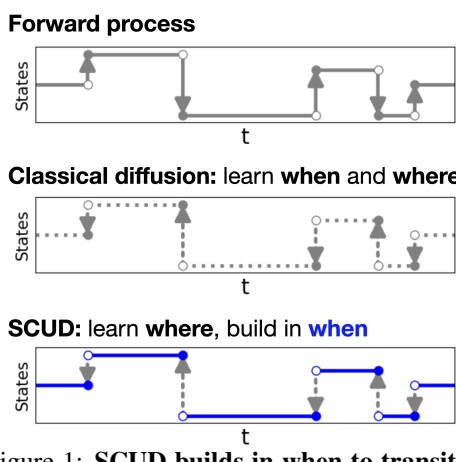

**Figure 1: SCUD builds in when to transition from the forward process, and only learns where to transition.**

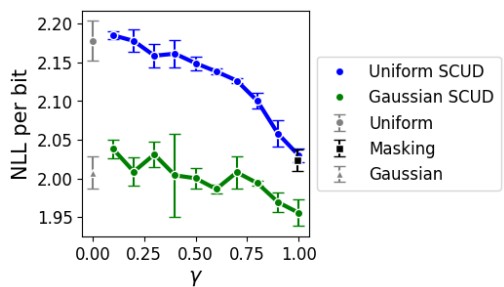

Figure 2: **Conditioning on the transition schedule results in a better fit to CIFAR10.** As $\gamma \to 1$, more information about the schedule is incorporated into the model. Models are fit on CIFAR10 with $B = 128$ states. We show mean and standard deviation over 3 replicates. Details are in App. D.

**Discrete diffusion** In diffusion, we start with a distribution that is easy to sample from, $q(x_1)$; we then learn a parameterized Markov process from time 1 to time 0 that evolves samples from $q(x_1)$ to a distribution $q_\theta(x_0)$ that is approximately $p(x_0)$. To learn a Markov process that evolves $q(x_1)$ to $p(x_0)$, we first pick a simple Markov process that approximately evolves samples from $p(x_0)$ to $q(x_1)$ from time 0 to 1; then we try to match the trajectories from the parameterized Markov process $q_\theta((x_t)_{t\in[0,1]})$ that evolves "backward" from time 1 to 0 to those of the simple process $p((x_t)_{t\in[0,1]})$ that evolve "forward" from time 0 to 1 (Campbell et al., 2022). We do so by maximizing the evidence lower bound (ELBO)

$$
\begin{aligned}
E_{p(x_0)} \log q_\theta(x_0) \geq & E_{p(x_0)} E_{p((x_t)_{t\in[0,1]}|x_0)} \log \frac{q_\theta((x_t)_{t\in[0,1]})}{p((x_t)_{t\in[0,1]}|x_0)} \\
= & E_{p((x_t)_{t\in[0,1]})} \log \frac{q_\theta((x_t)_{t\in[0,1]}|x_1)}{p((x_t)_{t\in[0,1]}|x_0,x_1)} + E_{p(x_1,x_0)} \log \frac{q(x_1)}{p(x_1|x_0)}.
\end{aligned}
\tag{1}
$$

This ELBO is maximized when the distribution of forward and backward trajectories match. The second term of the right hand side measures if the forward process indeed evolves samples $x_0 \sim p(x_0)$ to $q(x_1)$. The first term measures how well the forward and backward trajectories match.

**Discrete Markov processes and infinitesimal generators** To define a diffusion model, we need to define a simple Markov process to generate $p((x_t)_{t\in[0,1]})$ and we need to parameterize the backward Markov process Fortunately, discrete Markov processes are much easier to define than their continuous counterparts. Every time-homogeneous discrete Markov process is fully described by a $B \times B$ matrix that describes the "flow" of a particle at each instant in time known as the infinitesimal generator $\mathcal{L}$. In particular, $\mathcal{L}_{b,b'}$ describes the rate at which state $b$ transitions to state $b'$; the diagonal of $\mathcal{L}$ describes the rate of transitions out of $b$: $\mathcal{L}_{b,b} = -\sum_{b'\neq b} \mathcal{L}_{b,b'}$. Therefore, to simulate from a Markov process described by $\mathcal{L}$, starting at $x_t$, one simulates the time at which $x_t$ would transition to each other state $\Delta t_b \sim \text{Exp}(\mathcal{L}_{x_t,b})$ for $b \neq x_t$; then one transitions $x_t$ according to the first transition sampled: it take $\Delta t = \min_b \Delta t_b$ time to transition and $x_t$ transitions to $x_{t+\Delta t} = \text{argmin}_b \Delta t_b$. By a property of exponential distributions, the transition time is distributed according to the value on the diagonal of $\mathcal{L}$: $\Delta t \sim \text{Exp}(\sum_{b\neq x_0} \mathcal{L}_{x_0,b}) = \text{Exp}(-\mathcal{L}_{x_0,x_0})$. This procedure is known as the Gillespie algorithm (Gillespie, 1977).

**Picking the forward process** Two popular choices for the forward process are the uniform and masking processes. The uniform process has a constant rate of transitioning to any state ($\mathcal{L}_{b,b'} = 1/(B-1)$ if $b \neq b'$ and $\mathcal{L}_{b,b} = -1$) and the masking distribution has a constant rate of transition to a masking state $\emptyset$ (for $b \neq \emptyset$, $\mathcal{L}_{b,b'} = 0$ if $b \neq b' \neq \emptyset$, $\mathcal{L}_{b,\emptyset} = 1$, $\mathcal{L}_{b,b} = -1$, and $\mathcal{L}_{\emptyset,\emptyset} = 0$). Both of these processes are simple to simulate – simply sample $\Delta t \sim \text{Exp}(1)$ and then transition to a uniformly random state or to $\emptyset$. There are also other processes that bake in inductive biases for text, images, and proteins (Austin et al., 2021; Alamdari et al., 2023).

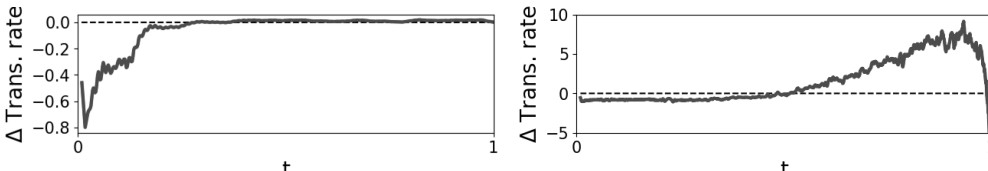

(a) D3PM (uniform forward process) on UniRef50.  (b) $\tau$LDR (Gaussian forward process) on CIFAR10.

Figure 3: **State of the art discrete diffusion models have backwards processes which do not match the forward process in when they transition.** We plot the transition rate of the backward process minus that of the forward process. We discuss details in App. D.

For typical Markov processes, information about the starting state $x_0$ becomes lost as $t$ gets larger and $p(x_t)$ gets closer to a stationary distribution $p(x_\infty)$. This distribution is a natural choice for $q(x_1)$ as long as $p(x_1|x_0)$ is close to converging to the stationary distribution.

In practice, $p(x_1|x_0)$ is usually not near $p(x_\infty)$, so we modulate the speed of the process by a rate $\beta_t$ at time $t$ – at the instant $t$ we simulate from the process $\beta_t \mathcal{L}$. Simulating this modulated process for time $t$ is equivalent to simulating the original process for time $\int_0^t \beta_s ds$. By choosing $\beta_t$ to become large as $t \to 1$, we can be sure $p(x_1|x_0) \approx p(x_\infty) = q(x_1)$.

**Parameterizing the backward distribution**  The backward Markov process is usually defined in terms of a parameterized, time-dependent, infinitesimal generator $\mathcal{L}_{\theta,t}$. The first term of Eqn. 1 is usually written as an integral in time $E_{t\sim\mathrm{Unif}(0,1)}L(\mathcal{L}_{\theta,t}, t)$, for some $L$ which intuitively measures how well the $\mathcal{L}_{\theta,t}$ describes the "flow" of the reversal of $p((x_t)_t)$ at instant $t$ (Campbell et al., 2022; Luo et al., 2022).

## 3  LEARNING WHEN AND WHERE TO TRANSITION

To fit a discrete diffusion model, the backward process should match the forward in both *when* it transitions and *where* it transitions to. One should expect that learning where to transition is hard; on the other hand, since the distribution of when to transition is simple and known a priori in many cases, one should expect learning when to transition should be trivial. We see however in Fig. 3 that this is not necessarily true – state of the art published diffusion models have detectable differences in the transition rates of their forward and backward processes.

Unlike previously derived forms of the ELBO which are written as an integral of the discrepancy of the flow at each moment $t$, with some algebra, we break up the ELBO into discrepancy of when and where to transition. Define the "transition schedule", $S = \{t_1, t_2, \ldots, t_M\}$, as the set of times at which $x_t$ transitions.

**Proposition 3.1.** *(Proof in Prop A.1 in the Appendix) The expression in Eqn. 1 is equal to the expression in Eqn 2 for some constant $C$.*

$$E_{p((x_t)_t)} \log \frac{q_\theta((x_t)_t|x_1,S)}{p((x_t)_t|x_0,x_1,S)} - \mathrm{KL}(p(S)||q_\theta(S)) - E_{p(S,x_0)}\mathrm{KL}(p(x_1|S,x_0)||q_\theta(x_1|S)) + C. \quad (2)$$

The first term represents the difference in log likelihoods between $q_\theta$ and $p$ when the transitions are known – it measures if the forward and backward processes match where they transition to. The second term measures if the forward and backward processes match when they transition. The third term, like the second term of Eqn. 1 intuitively measures if $p(x_1|x_0)$ has converged to $p(x_\infty)$.

To build diffusion models that better fit their objective, we therefore would like to incorporate knowledge of $p(S)$ into the model. Eqn 2 is suggestive of how to do this: set $q(S) = p(S)$ so that the second term becomes $0$ and then learn where to transition by optimizing the first term. We call this procedure "schedule conditioning" (Fig. 1) and in Sec. 4 we describe how to perform it in practice.

Unlike diffusion models with the uniform forward process, diffusion models with the masking forward process are parameterized so that the distribution of times at which tokens are masked matches the distribution of times at which they are unmasked – these models know when to transition. In practice they have been observed to outperform uniform diffusion models. In Sec. 5 we will prove

that applying our methods in Sec. 4 gives exactly masking distribution, explaining their superior performance. By schedule conditioning other processes with more appropriate inductive biases, we also improve on masking diffusion (Fig 2).

# 4 SCHEDULED CONDITIONED DIFFUSION (SCUD)

In this section, motivated by Eqn. 2, we describe how to incorporate information about when to transition into a discrete diffusion model. Ideally we could set $q(S) = p(S)$; however, in general, $\mathcal{L}$ may not have constant transition rates at each state, in which case $S$ may be correlated with $x_0$ and $p(S)$ may be a complex distribution. Instead of looking directly at transitions then, we introduce latent "events" which will act as transitions did above – they occur with constant rate and often result in transitions; in some cases we discuss below, they will coincide exactly with transitions. $S$ will describe the schedule of these events and this is what we'll condition on.

In Sec. 4.1 we will describe models that condition on these event schedules, SCUD. Next in Sec. 4.2 we will write the loss in a form that is easy to train on high dimensional data. Finally in Sec. 4.3 we will describe how to parameterize and sample from SCUD.

## 4.1 CONDITIONING ON EVENT SCHEDULES

**Markov processes with event schedules**   To sample from a uniform forward process starting at $x_t$, we sampled a transition time according to a rate that was independent of the current state, $\Delta t \sim \text{Exp}(1)$, and then sampled $x_{t+\Delta t}$ with uniform probability. Consider more generally the discrete Markov process on $x_t$ such that we sample an "event" $\Delta t \sim \text{Exp}(r)$, and then sample $x_{t+\Delta t} \sim \text{Categorical}(K_{x_t, \cdot})$ where $K_{x_t, \cdot}$ is a matrix whose rows are normalized distributions; note in this case $x_t$ may be equal to $x_{t+\Delta t}$. By appealing to the formal definition of $\mathcal{L}$, the next proposition tells us that this process has infinitesimal generator that flows according to the rate $r \times K$, with a $-I$ to describe the flow out of $x$.

**Proposition 4.1.** *(Proof in Prop A.2 in the Appendix) The infinitesimal generator of this process is $\mathcal{L} = r(K - I)$ where $I$ is the identity matrix. In particular, any Markov process with $\mathcal{L}$ can be simulated in the above way by picking an $r \geq \max_b -\mathcal{L}_{b,b}$ and setting $K = \mathcal{L}/r + I$.*

We note there are many choices of $r$ that allow one to write the same Markov process in this way and we will evaluate different choices in Sec. 5.

**Reversing the process conditioned on the event schedule**   Call $p((x_t)_t)$ the distribution of paths that start at $p(x_0)$ and evolve according to the above Markov process. The next proposition uses a bit of algebra to suggest that we can simulate from $p((x_t)_t)$ "backwards" by 1) sampling the ending point $x_1 \sim p(x_1)$, 2) sampling the event schedule $\{t_1, t_2, \ldots, t_M\} \sim p(S)$, and then 3) going backwards, sampling where the particle came from at the $m$-th event.

**Proposition 4.2.** *(Proof in Prop A.3 in the Appendix) Call the event schedule $S = \{t_1, t_2, \ldots, t_M\}$ and $t_0 = 0$. Call $s_t$ the number of events up to time $t$, so $s_{t_m} = m$.*

$$p((x_t)_t, S) = p(S)p(x_1) \prod_{m=1}^{M} p(x_{t_{m-1}} | x_{t_m}, s_{t_m}). \tag{3}$$

We now aim to model this backwards process.

**SCUD: schedule conditioned discrete diffusion models**   As suggested in Sec 3, we wish to build a discrete diffusion model $q_\theta$ by setting $q(x_1) = p(x_\infty)$ and $q(S) = p(S)$. Prop. 4.2 suggests parameterizing $q$ so that, at each event, it predicts the previous state $x_{t_{m-1}}$ given 1) the current state $x_{t_m}$ and 2) the number of events that have occurred so far $s_t$. We call such a model a SCUD (schedule conditioned diffusion) model. With some algebra, in analogy with Eqn. 2 we get a closed form objective.

**Proposition 4.3.** *(Proof in Prop A.4 in the Appendix) Calling the event schedule $S = \{t_1, t_2, \ldots, t_M\}$ and $t_0 = 0$,*

$$
\begin{aligned}
E_{p(x_0)} \log q_\theta(x_0) \geq & E_{p((x_t)_t, S, x_0)} \sum_{m=1}^{M} \text{KL}(p(x_{t_{m-1}} | x_{t_m}, x_0, s_{t_m}) || q_\theta(x_{t_{m-1}} | x_{t_m}, s_{t_m})) \\
& - E_{p(S, x_0)} \text{KL}(p(x_1 | s_1, x_0) || p(x_\infty)).
\end{aligned}
\tag{4}
$$

*This objective is minimized when $q_\theta(x_{t_{m-1}} | x_{t_m}, s_{t_m}) = p(x_{t_{m-1}} | x_{t_m}, s_{t_m})$.*

The first term is from the first term of Eqn 2 and teaches $q_\theta$ where to go at each event. The second term of Eqn 2 vanishes and the third term becomes the third term of Eqn 4, which should be small if $p(x_1)$ converges to $p(x_\infty)$. By Prop. 4.2 then, as the objective in Eqn. 4 is minimized, $q_\theta((x_t)_t)$ approaches $p((x_t)_t)$.

**Computing the objective** The ELBO in Eqn. 4 is straightforward to compute. To calculate the first term, we note, writing each state as a one-hot vector,

$$p(x_{t_{m-1}}|x_{t_m}, x_0, s_t) = \frac{p(x_{t_{m-1}}|x_0, s_t)p(x_{t_m}|x_{t_{m-1}}, s_t)}{p(x_{t_m}|x_0, S)} = \frac{x_0^T K^{s_t-1} x_{t_{m-1}} x_{t_{m-1}}^T K x_{t_m}}{x_0^T K^{s_t} x_{t_m}}. \quad (5)$$

To calculate the second, we note $p(x_\infty)$ can be derived as the left eigenvector of $\mathcal{L}$ that corresponds to the eigenvalue 0 (as it does not change under flow from $\mathcal{L}$) and $p(x_1|s_1, x_0) = x_0^T K^{s_1} x_1$.

## 4.2 HIGH DIMENSIONAL DATA

For high dimensional discrete data such as images, language, and biological sequences, it is common to choose processes $\mathcal{L}$ that act on each dimension independently. Say our data is $D$ dimensional with dimensions $x_0^1, \ldots, x_0^D$ with each $x_0^d$ a discrete object in a set of size $B$. We extend SCUD to this case by simulating $D$ parallel schedules for each dimension $S^1, \ldots, S^D \sim p(S)$; here $s_t$ becomes a $D$-dimensional vector.

**Parameterizing $q_\theta$** For a time $t$, if $s_t^d > 0$, define $\text{pr}(x_t^d)$ as the state at the last event in dimension $d$ and $\text{pr}(x_t)$ the previous state at each dimension; i.e. if the event schedule at dimension $d$ is $S^d = \{t_1^d, \ldots, t_m^d\}$ and $t \in [t_m^d, t_{m+1}^d)$, then $\text{pr}(x_t^d) = x_{t_{m-1}^d}^d$. Our formula for reversing $p((x_t)_t)$ in Prop. 4.2 remains the same, but in App. B.2 we show $p(\text{pr}(x_t)|x_t, s_t)$ factorizes. Thus we parameterize our predictor $q_\theta(\text{pr}(x_t)|x_t, s_t)$ so it also factorizes as $\prod_{d=1}^D q_\theta(\text{pr}(x_t^d)|x_t, s_t)$. Thus we get an objective as in Eqn 4 but with a sum over $D$ in front.

**Efficient loss** We could technically use our objective in Eqn. 4 by taking empirical estimates of the expectation and the sum over events. In this case however, each empirical sample corresponds to one event which effects a single dimension $d$, so it only checks the prediction $q_\theta(\text{pr}(x_t^d)|x_t, s_t)$. The loss of other diffusion models, written as $E_{t \sim \text{Unif}(0,1)} E_{x_t \sim p(x_t|x_0)} \sum_{d=1}^D L^d(\mathcal{L}_\theta, x_t, t|x_0, \mathcal{L})$, allow one to sample $t$ and then check the predictions of $q_\theta(\text{pr}(x_t^d)|x_t, s_t)$ at that time for every $d$ in parallel. To write our objective in a similar form, we sample $t \sim \text{Unif}(0, 1)$ and then add a weight $s_t^d \times \beta_t / \int_0^t \beta_s ds$ representing how likely an event is to occur at the instant $t$:

**Proposition 4.4.** *(SCUD loss) (Proof in Prop. A.6 in the Appendix) The first term of Eqn. 4 is*

$$-E_{t \sim \text{Unif}(0,1)} E_{p(x_t, x_0, S)} \frac{\beta_t}{\int_0^t \beta_s ds} \sum_d s_t^d \text{KL}(p(\text{pr}(x_t^d)|x_t^d, s_t^d, x_0^d)||q_\theta(\text{pr}(x_t^d)|x_t, s_t)). \quad (6)$$

We can approximate this objective by empirical estimates of all of the expectations and optimize with minibatch gradient descent. For a single evaluation of $q_\theta$ we can predict $\text{pr}(x_t^d)$ for each dimension $d$ in parallel and check whether it matches the forward process along every dimension. The algorithm for calculating an estimate of the ELBO for a $x_0$ is summarized in App. B.1.

## 4.3 SCHEDULE CONDITIONING IN PRACTICE

**Parameterization** $q_\theta$ must predict, for each dimension, $p(\text{pr}(x_t^d)|x_t, s_t)$, which is an expectation over the posterior of $x_0^d$ given $x_t$ and $S$:

$$\sum_{x_0^d} p(\text{pr}(x_t^d)|x_t^d, s_t^d, x_0^d)p(x_0^d|x_t, S) = \sum_{x_0^d} p(x_t^d|\text{pr}(x_t^d))p(\text{pr}(x_t^d)|s_t^d, x_0^d) \frac{p(x_0^d|x_t, s_t)}{p(x_t^d|s_t^d, x_0^d)}.$$

In App. B.2 we show that the fraction on the right hand side is proportional to $p(x_0^d|x_t^{-d}, s_t^{-d})$ where $x_t^{-d}$ and $s_t^{-d}$ are $x_t$ and $s_t$ without dimension $d$. Other discrete diffusion methods parameterize their $q_\theta$ to predict analogues of this quantity – Austin et al. (2021) predicted a similar quantity rather than

directly predicting $p(x_0^d|x_t, S)$, and predicting $p(x_0^d|x_t^{-d}, s_t^{-d})$ is identical to predicting $p(x_0^d|x_t, S)$ when $x_t^d$ is masked. Predicting this quantity has the benefit that we do not need to learn what $x_t^d$ tells us about $x_0^d$; it is rather baked into our prediction. We parameterize our $q_\theta$ similarly.

Thus, to predict $q_\theta(\text{pr}(x_t^d)|x_t, S)$ we input $x_t$ and $s_t$ into a neural network that outputs a vector of probabilities $\tilde{x}_{0,\theta}$ and set

$$q_\theta(\text{pr}(x_t^d)|x_t, s_t) = \sum_b p(x_t^d|\text{pr}(x_t^d))p(\text{pr}(x_t^d)|s_t^d, x_0^d = b)\tilde{x}_{0,\theta,b} = Kx_t^d \circ K^{s_t^d-1,T}\tilde{x}_{0,\theta}. \quad (7)$$

Note we do not explicitly forbid $\tilde{x}_{0,\theta}$ from using $x_t^d, s_t^d$ to predict $x_0^d$.

**Sampling** To sample, in principle we could take $x_1 \sim p(x_\infty)$, $S \sim p(S)$, and then iteratively reverse each event in $S$ in order using our predictions of $q_\theta(\text{pr}(x_t^d)|x_t, s_t)$. For data with many dimensions however, $S$ could contain tens of thousands of events, requiring many evaluations of $\tilde{x}_{0,\theta}$. Instead, like Campbell et al. (2022) and Zhao et al. (2024), we reverse many events at once. In particular we use an analogue of a k-Gillespie procedure (Zhao et al., 2024) – we pick $k$ events to reverse and reverse them with a single evaluation of $\tilde{x}_{0,\theta}$. We describe the particulars of which transitions to reverse and how to many transitions at once in App. B.3.

**Choosing the rate $\beta_t$** Our choice of $\beta_t$ describes how we compress the forward process running from time 0 to $\int_0^1 \beta_s ds$ into the interval $[0, 1]$. It controls what times we sample when training the objective Eqn. 6 and $\int_0^1 \beta_s ds$ controls the convergence of $p(x_1)$ to $p(x_\infty)$. Austin et al. (2021) suggest picking $\beta_t$ so that the mutual information between $x_0$ and $x_t$ decreases linearly to $\epsilon$ on the interval $[0, 1]$. For SCUD models, we pick $\beta_t$ so that the same is true when conditioning on the schedule: $E_{s_t}\text{MI}(x_0, x_t|s_t)$ decreases linearly on the interval $[0, 1]$. We discuss details in App. B.4.

## 5 SCHEDULE CONDITIONING TO CONDITION ON TRANSITIONS

To incorporate information about transitions into $q_\theta$, we wish to condition on the schedule. We described how conditioning on "events" in the previous section allow us to incorporate this structure. However not every event corresponds to a transition. The amount of information about the transitions that we bake into our model depends on the diagonal of $K$ – the probabilities of no transition at an event. In turn the diagonal of $K$ will depend on our choice of the rate of events $r$. For a fixed $\mathcal{L}$, we can choose any rate $r \geq r^* = \max_b -\mathcal{L}_{b,b}$. Let's parameterize our choices of rate with a parameter $\gamma$: let $r = \gamma^{-1}r^*$. When $\gamma$ is 1, the rate of events is as slow as possible; when $\gamma \to 0$, the rate of events goes to $\infty$.

$\gamma$ controls the diagonal of $K$ and therefore how much we condition on the schedule. We can write $K = \gamma\mathcal{L}/r^* + I$; the larger $\gamma$ is, the smaller the diagonal of $K$. When $\mathcal{L}$ is "normalized" so that every entry on the diagonal is the same, $\gamma = 1$ coincides with $K$ with zero diagonal; in this case, every event is a transition and we've fully conditioned on the transition schedule. On the other hand, as $\gamma \to 0$, the diagonals of $K$ get closer to 1, so that almost no events result in a transition.

We now show that when $\mathcal{L}$ is uniform and $\gamma = 1 - 1/D$, that is, we nearly fully condition on the schedule, our process is equivalent to masking diffusion. On the other hand, as $\gamma \to 0$, we learn a backwards process while baking in no information about transitions; we show this recovers classical discrete diffusion exactly.

### 5.1 CONNECTION TO MASKING DIFFUSION

Say $\gamma = 1 - 1/B$ and $\mathcal{L}$ is uniform: $\mathcal{L}_{b,b'}$ is $1/B$ when $b \neq b'$. For this choice, $K$ is a matrix which has $1/B$ at every position. If a token is corrupted at least once by $K$ then it is distributed uniformly; it tell us nothing about $x_0$ so it is as if that token is "masked". When we condition on the event schedule, $s_t$ will tell us exactly which positions are masked when $s_t^d > 0$. By integrating out $s_t$ conditioned on the mask, we get exactly the masking diffusion objective (Shi et al., 2024).

**Proposition 5.1.** *(Proof in Prop. A.7 in the Appendix) Call the masking indicator $m_t^d = s_t^d > 0$. $\tilde{x}_{0,\theta}(x_t, s_t)$ only depends on $s_t$ through $m_t$. Defining $\alpha_t = \exp(-\int_0^t \beta_s ds)$, the objective Eqn. 6 is*

$$E_{t\sim\text{Unif}(0,1)}E_{p(m_t)}E_{p(x_t|x_0,m_t)}\frac{\beta_t\alpha_t}{1-\alpha_t}\sum_d x_0^T \log\tilde{x}_{0,\theta}(x_t, m_t)^d.$$

*The mask $m_t$ is distributed according to $m_t^d \sim \text{Bern}(1 - \alpha_t)$.*

In App. B.4 we also show that our choice for rate $\beta_t$ discussed in Sec. 4.3 for this SCUD process is linear (in the sense $\alpha_t = 1 - t$), just as for the masking process as discussed in (Austin et al., 2021).

## 5.2 CONNECTION TO CLASSICAL DISCRETE DIFFUSION

As $\gamma \to 0$, each event represents an infinitesimal change in $x_t$. As well, the number of events up to time $t$, $s_t$, grows larger but fluctuates less and less; inputting $s_t$ into $q_\theta(\text{pr}(x_t^d)|x_t, s_t)$ becomes approximately identical to inputting the time $t$ into $q_\theta$. Therefore, as $\gamma \to 0$, $q_\theta$ predicts the infinitesimal change at time $t$: the infinitesimal generator. This is exactly the objective of classical discrete diffusion. The next proposition shows that when we take the limit $\gamma \to 0$ we recover exactly the loss from SEDD (Luo et al., 2022) which is also equivalent to that from $\tau$LDR (Campbell et al., 2022).

**Proposition 5.2.** *(Proof in Prop. A.8 in the Appendix) Define the score function estimator as in SEDD (Luo et al., 2022)[1]*

$$\tilde{s}(x_t, t)_{\theta,b}^d = \frac{q_\theta(x_t^d = b|x_t^{-d})}{q_\theta(x_t^d|x_t^{-d})} := \frac{E_{\tilde{x}_{0,\theta}(x_t, s_t)} p(x_t^d = b|x_0^d)}{E_{\tilde{x}_{0,\theta}(x_t, s_t)} p(x_t^d|x_0^d)}.$$

*Suppressing the dependence of $\tilde{s}_\theta$ on $x_t, t$, as $\gamma \to 0$ the objective in Eqn. 6 converges to*

$$-E_{t \sim \text{Unif}(0,1)} E_{p(x_0, x_t)} \beta_t \sum_d \left[ \sum_{b \neq x_t^d} \mathcal{L}_{b, x_t^d} \left( \tilde{s}_{\theta,b}^d - \frac{p(x_t^d = b|x_0^d)}{p(x_t^d|x_0^d)} \log \tilde{s}_{\theta,b}^d - g\left( \frac{p(x_t^d = b|x_0^d)}{p(x_t^d|x_0^d)} \right) \right) \right]$$

*where $g(x) = x(\log x - 1)$.*

In App. B.4 we also show that our choice for rate $\beta_t$ discussed in Sec. 4.3 approaches the rate function for classical discrete diffusion as $\gamma \to 0$.

## 6 RELATED WORK

Diffusion generative models are state of the art for images and other continuous data (Ho et al., 2020; Dhariwal & Nichol, 2021; Peebles & Xie, 2022), but have so far lagged behind autoregressive models on discrete sequence data like text. Inspired by its success on continuous modalities, a number of works have attempted to extend diffusion to discrete domains. D3PM (Austin et al., 2021), for example, adapts Ho et al. (2020)'s continuous framework and extends early work by Hoogeboom et al. (2021) by introducing a family of categorical noise processes based on structured discrete transition matrices. Our method takes inspiration from the diverse noise processes explored in D3PM but is ultimately more flexible, as our formalism can use any $\mathcal{L}$ which converges to a stationary distribution and does not require doubly stochastic matrices.

To allow for more flexible sampling and principled model development, a number of methods have extended diffusion from discrete time to continuous time. For example, $\tau$LDR (Campbell et al., 2022) intro a continuous-time Markov chain formulation and a corresponding continuous-time ELBO. In related work, SEDD (Lou et al., 2023) introduced score-matching loss for discrete spaces, intended to parallel score-matching for continuous spaces (Song & Ermon, 2019), which allows flexible continuous time sampling. These models differ primarily in how they are parameterized and how they estimate the ELBO objective. SCUD on the other hand is more flexible as it only requires that one can calculate matrix vector products with $K$, or equivalently $\mathcal{L}$. Recently, many works have chosen to focus purely on masking state diffusion, proposed weighted losses that have pushed compression metrics closer and closer to numbers obtained from autoregressive models (Sahoo et al., 2024; Shi et al., 2024; Ou et al., 2024). While SCUD is closely related to recent work in continuous-time discrete diffusion, we find that schedule conditioning allows structured noise processes to improve performance and thereby leads to non-masking diffusion with state-of-the-art performance.

In the realm of sampling, Chen et al. (2023) also considered an accelerated procedure for simple diffusion models in which the transition schedule is sample sampled first followed by the transitions conditioned on the schedule, which shares similar motivations with SCUD. SCUD, however, describes how to build in this information into training a model.

---

[1] recall $\tilde{x}_{0,\theta}^d(x_t, s_t)$ is trained to fit $p(x_0^d|x_t^{-d}, s_t^{-d})$.

Lastly, inspired by flow-matching developments in image modeling, many authors have begun to propose flow-matching frameworks for discrete data. Campbell et al. (2024b) for instance propose a flow-matching framework that accommodates joint modeling of discrete and continuous modalities, enabling applications in protein design. Similarly, Gat et al. (2024) presents a general framework for learning probability paths on discrete sequences and trains large-scale models on text datasets. Unlike papers on discrete flow matching, we still employ a diffusion framework and use an ELBO loss, but it's possible that our investigation of schedule conditioning or structured forward processes could yield insights that are also useful for score matching, as many of the underlying modeling methods are shared.

## 7 RESULTS

We show that by incorporating information about transitions, SCUD better fits the forward process. We first demonstrate the results of Sec. 5 that SCUD with a uniform forward process interpolates between uniform and masking discrete diffusion. We next show that applying SCUD to state of the art classical discrete diffusion models without schedule conditioning improves their likelihoods on images, text, and protein data. Finally, by building SCUD with forward processes that build in inductive biases, we also show scale that we can improve over SCUD uniform which is similar to masking (Sec. 5.1), thereby unlocking the potential of structured discrete diffusion. Throughout this section, SCUD refers to $\gamma = 1$.

The structured forward processes we build for each modality will be inspired by those from Austin et al. (2021). However Austin et al. (2021) used processes in discretized time that are not equivalent to and continuous time Markov process; thus we describe new structured processes for continuous time in terms of $\mathcal{L}$ or $K$.

In all cases we try to make only minor modifications to the architecture and training parameters from previous models so that differences in scores are due to schedule conditioning. We employed a few strategies so that moving from classical discrete diffusion to SCUD did not add substantial computational overhead, summarized in App. D.5. Other experimental details are in App. D.

### 7.1 CONNECTION TO OTHER MODELS

Here we show that by incorporating information about the distribution of transitions into a discrete diffusion model, one gets better fits to the forward process.

We fit models to CIFAR10 where each pixel takes a value from 1 to $B = 128$. In Fig. 2 we see that on this dataset discrete diffusion with a uniform forward process is outperformed by masking diffusion. We see that sweeping $\gamma$ between 0.1 and 1, SCUD with the uniform forward process interpolates the performance of the two models as predicted above.

Next we build a structured forward process that builds in the inductive bias that similar pixel values describe similar colors – we set $\mathcal{L}_{i,j} = \exp\left(-200\left(\frac{i-j}{B}\right)^2\right)$, similar to the discrete-time Gaussian forward process in Austin et al. (2021). We see that a discrete diffusion model with this forward process slightly outperforms masking distribution. We next build SCUD models with this forward process; we see that these models better fit their objective as we incorporate more information about transitions – $\gamma \to 1$. These models outperform models that have structured forward processes (Gaussian) or those that just condition on the transition schedule (masking) without doing the other.

### 7.2 IMAGES

Here we build models on CIFAR10 with $B = 256$ and compare to state of the art diffusion models. We use the architecture from (Kingma et al., 2021) as in discrete diffusion models MD4 (Shi et al., 2024) and similar to that in D3PM (Austin et al., 2021) and $\tau$LDR Campbell et al. (2022). To incorporate $s_t$ into our function, we replace additive layers that inject $t$ into every layer with FiLM (Perez et al., 2017) layers that incorporate $s_t$ into every layer. We also use the logistic parameterization from Salimans et al. (2017) also used in D3PM, which interprets the output of the model as the parameters of a discretized logistic distribution over pixel values, so that similar pixel intensities have similar probabilities.

| Method | Forward process | Training samples | BPD |
|--------|----------------|------------------|-----|
| D3PM | Uniform | $1.9 \times 10^8$ | 5.08 |
| D3PM | Gaussian | $1.9 \times 10^8$ | 3.44 |
| $\tau$LDR | Gaussian | $2.6 \times 10^8$ | 3.59 |
| MD4 | Masking | $2.6 \times 10^8$ | **2.78** |
| Classical | Gaussian | $6.4 \times 10^7$ | 2.94 |
| Masking | Masking | $6.4 \times 10^7$ | 2.90 |
| SCUD | Gaussian | $6.4 \times 10^7$ | *2.86* |

Table 1: **Schedule conditioning improves model fit on images.** We compare to other discrete diffusion models and report model fit in bits per dimension on CIFAR10. Models labelled "Gaussian" implement numerically different forward processes that are united in a Gaussianity assumption.

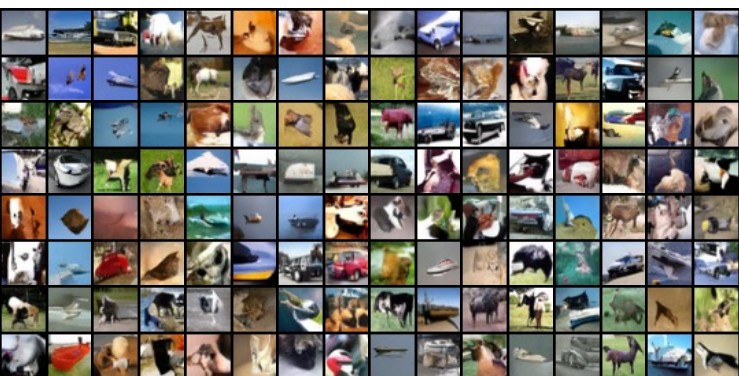

Figure 4: **Samples from SCUD Gaussian trained on CIFAR10.**

In Table 1 we compare SCUD with discrete diffusion models D3PM (Austin et al., 2021), $\tau$LDR (Campbell et al., 2022), and MD4 (Shi et al., 2024) as well as our implementations of classical discrete diffusion models. We see that applying SCUD to model the Gaussian forward processes substantially improves likelihood with a fraction of the compute. Among previous discrete diffusion models, masking diffusion is the most performant despite not incorporating inductive biases. When controlled for compute in our baselines, SCUD beats masking. This suggests that masking beats Gaussian diffusion in classical models because the benefit of schedule conditioning outweighs the benefit of incorporating inductive biases. By both incorporating inductive biases and schedule condition, SCUD unlocks the potential of Gaussian discrete diffusion on images.

Fig. 4 shows samples from SCUD Gaussian. The samples from SCUD resemble real objects much more than those from autoregressive models PixelCNN++ (Salimans et al., 2017) and PixelSNAIL (Chen et al., 2017) which have state of the art likelihoods. However they do not contain clear objects like those from D3PM (Austin et al., 2021) or $\tau$LDR (Campbell et al., 2022); MD4 did not show or evaluate images. The quality of samples from those models are known to depend heavily on modelling choices, such as modifications of the objective, choice of $\beta_t$, and training time; and sampling procedure, such as the inclusion of corrector steps or how to denoise many dimensions at once. Here we focus on achieving low likelihoods and leave the task of translating a better fit into higher quality images to future work.

### 7.3 LANGUAGE

Here we build models on the one billion words dataset with a $B = 30522$ vocabulary size. To improve over masking diffusion, we want to build in inductive biases about which vocabulary tokens are more similar. However, it is not trivial to efficiently simulate a process over 30 thousand states. To do so, we define a sparse 10-nearest neighbour graph over the most frequent 2000 states, which make up 95% of tokens in the data. Our forward process diffuses along this graph with some probability or transitions approximately uniformly with some small probability; the less frequently used 25 thousand states always transition uniformly. We discuss the details in App C.

| Method | Forward process | Training tokens | Perplexity |
|--------|----------------|-----------------|------------|
| SEDD | Uniform | $3.3 \times 10^{10}$ | 40.25 |
| SEDD | Masking | $3.3 \times 10^{10}$ | 32.79 |
| MDLM | Masking | $3.3 \times 10^{10}$ | **27.04** |
| SCUD | Uniform | $1.1 \times 10^{10}$ | 37.82 |
| SCUD | Nearest Neighbour | $1.1 \times 10^{10}$ | *37.63* |

Table 2: **Schedule conditioning improves model fit on language.** We compare to other discrete diffusion models on LM1B.

| Method | Forward process | Training tokens | Perplexity |
|--------|----------------|-----------------|------------|
| D3PM | Uniform | $> 3 \times 10^{11}$ | 18.82 |
| D3PM | BLOSUM | $> 3 \times 10^{11}$ | 17.16 |
| D3PM | Masking | $> 3 \times 10^{11}$ | **14.61** |
| Classical | BLOSUM | $8 \times 10^{9}$ | 15.39 |
| Masking | Masking | $8 \times 10^{9}$ | 15.56 |
| SCUD | BLOSUM | $8 \times 10^{9}$ | *15.29* |

Table 3: **Schedule conditioning improves model fit on proteins.** We implement and compare to the small architecture from (Alamdari et al., 2023) on UniRef50.

SCUD allows one to flexibly incorporate a forward process by only requiring one to define $K$ and take powers to evaluate likelihoods. Classical discrete diffusion models such as SEDD on the other hand require closed form $p(x_t|x_0)$ which requires a matrix exponential to evaluate. While in some cases the matrix exponential is easy to evaluate, that is not the case for our forward process. This also means that we could not compare to classical diffusion on this structured classical diffusion.

In Tab. 2 we compare SEDD (Luo et al., 2022) and MDLM (Sahoo et al., 2024) to SCUD and an ablation without structure, SCUD uniform. As expected, among previous models, masking beats uniform; in (Austin et al., 2021) it was noted that masking also beats discrete diffusion with a nearest neighbour structure on this dataset[2]. We see again that applying SCUD to uniform diffusion improves its fit to the data with a fraction of the compute. We also again see that unlike previous discrete diffusion models, when we add structure to the forward process, we improve our fit.

### 7.4 PROTEINS

Here we train models on the UniRef50 protein dataset with architectures from (Alamdari et al., 2023). As in (Alamdari et al., 2023) we build a forward process using the BLOSUM matrix; this matrix describes the rates of mutations between amino acids seen in nature. We describe the details of our process in App C; we note $B = 31 = 20$ canonical amino acids $+ 11$ special tokens.

In Tab. 3 we compare SCUD BLOSUM with the small D3PM models from (Alamdari et al., 2023) as well as our implementations of classical discrete diffusion models. We see again that applying SCUD to uniform and BLOSUM diffusion substantially improves the model fit given a fraction of the compute budget. In classical discrete diffusion, masking strongly outperforms BLOSUM diffusion. We see the opposite for SCUD, where by both schedule conditioning and incorporating inductive biases, SCUD BLOSUM outperforms masking, and thereby unlocks the potential of BLOSUM diffusion.

## 8 CONCLUSION

The choice of forward process is critical to the definition of a discrete diffusion model. Yet previous results have shown very strong performance from the simplest forward process – the masking process. SCUD offers an explanation for the superior performance of masking diffusion – it incorporates information about the transition schedule. By incorporating this information into models with other forward processes, SCUD allows us to build models that build in inductive biases and outperform masking.

---

[2]These models achieved much worse perplexity values than the models in Tab. 2 but are not directly comparable due to a different choice of tokenizer

## 9 REPRODUCIBILITY

We include code to train, evaluate, and sample from SCUD models in our code release. We include implementations for the exact architectures used in our experiments. The training and evaluation details for experiments we ran on images, language and proteins were described by previous papers and again in our appendix.

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

## A  PROOFS OF RESULTS

**Proposition A.1.** *(Proof of Prop 3.1) The expression in Eqn. 1 is equal to the expression in Eqn 2 for some constant $C$.*

*Proof.* $S$ is a deterministic function of $(x_t)_t$ so we can write the first term of Eqn. 1 as

$$
\begin{aligned}
E_{p((x_t)_{t\in[0,1]}|x_0)} \log \frac{q_\theta((x_t)_{t\in[0,1]}|x_1)}{p((x_t)_{t\in[0,1]}|x_0,x_1)} =& E_{p((x_t)_{t\in[0,1]}|x_0)} \log \frac{q_\theta((x_t)_{t\in[0,1]},S|x_1)}{p((x_t)_{t\in[0,1]},S|x_0,x_1)} \\
=& E_{p((x_t)_{t\in[0,1]}|x_0)} \log \frac{q_\theta((x_t)_{t\in[0,1]}|x_1,S)}{p((x_t)_{t\in[0,1]}|x_0,x_1,S)}. \\
& + E_{p(S,x_1|x_0)} \log \frac{q_\theta(S|x_1)}{p(S|x_0,x_1)}.
\end{aligned}
\tag{8}
$$

We can combine the second term of this equation with the second term of Eqn. 1 to get

$$E_{p(S,x_1|x_0)} \log \frac{q_\theta(S|x_1)}{p(S|x_0,x_1)} + E_{p(x_1|x_0)} \log \frac{q(x_1)}{p(x_1|x_0)}$$

$$= E_{p(S|x_0)} \log \frac{q_\theta(S)}{p(S|x_0)} + E_{p(S,x_1|x_0)} \log \frac{q_\theta(x_1|S)}{p(x_1|x_0,S)}$$

$$= E_{p(S|x_0)} \log \frac{q_\theta(S)}{p(S)} + E_{p(S|x_0)} \log \frac{p(S)}{p(S|x_0)} - E_{p(S|x_0)} \mathrm{KL}(p(x_1|x_0,S)|q_\theta(x_1|S)).$$

$$(9)$$

The first term is $-\mathrm{KL}(p(S)||q_\theta(S))$ and the second does not depend on $q$. This completes the proof. $\qquad\square$

**Proposition A.2.** *(Proof of Prop 4.1) The infinitesimal generator of this process is $\mathcal{L} = r(K - I)$ where $I$ is the identity matrix. In particular, any Markov process with $\mathcal{L}$ can be simulated in the above way by picking an $r \geq \max_b -\mathcal{L}_{b,b}$ and setting $K = \mathcal{L}/r + I$.*

*Proof.* The process is described is clearly Markov. By the formal definition of $\mathcal{L}$, for $b' \neq b$,

$$\mathcal{L}_{b,b'} = \lim_{t\to 0} \frac{1}{t} p(x_t = b'|x_0 = b)$$

$$= \lim_{t\to 0} \frac{1}{t} (p(\text{an event occurs before } t) \times p(\text{the event transitions to } b') + o(t)) \qquad (10)$$

$$= \lim_{t\to 0} \frac{1}{t} (1 - e^{-rt}) K_{b,b'} = r K_{b,b'}.$$

Then, since the rows of $K$ sum to 1,

$$\mathcal{L}_{b,b} = -\sum_{b'\neq b} \mathcal{L}_{b,b'} = -r \sum_{b'\neq b} K_{b,b'} = -r(1 - K_{b,b}).$$

The second statement follows from rearranging the first. The requirement that $r \geq \max_b -\mathcal{L}_{b,b}$ comes from the fact that all entries in $K$ must be non-negative and $K_{b,b} = \mathcal{L}_{b,b}/r + 1$. $\qquad\square$

**Proposition A.3.** *(Proof of Prop 4.2 in the Appendix) Call the event schedule $S = \{t_1, t_2, \ldots, t_M\}$ and $t_0 = 0$. Call $s_t$ the number of events up to time $t$, so $s_{t_m} = m$.*

$$p((x_t)_t, S) = p(S)p(x_1) \prod_{m=1}^{M} p(x_{t_{m-1}}|x_{t_m}, s_{t_m}). \qquad (11)$$

*Proof.*

$$p((x_t)_t, S) = p(S)p(x_1)p(x_{t_{0:M}}|x_1, S)$$

$$= p(S)p(x_1) \prod_{m=1}^{M} p(x_{t_{m-1}}|x_{t_{m:M}}, S).$$

By the Markov property, $p(x_{t_{m-1}}|x_{t_{m:M}}, S) = p(x_{t_{m-1}}|x_{t_m}, S)$. Finally, $p(x_{t_{m-1}}|x_{t_m}, S) \propto p(x_{t_m}|x_{t_{m-1}}, S)p(x_{t_{m-1}}|S) = p(x_{t_m}|x_{t_{m-1}})p(x_{t_{m-1}}|s_{t_{m-1}})$ only depends on $S$ through $s_{t_{m-1}}$, or equivalently, $s_{t_m} = 1 + s_{t_{m-1}}$. $\qquad\square$

**Proposition A.4.** *(Proof of Prop. 4.3) Calling the event schedule $S = \{t_1, t_2, \ldots, t_M\}$ and $t_0 = 0$.*

$$E_{p(x_0)} \log q_\theta(x_0) \geq - E_{p((x_t)_t, S, x_0)} \sum_{m=1}^{M} \mathrm{KL}(p(x_{t_{m-1}}|x_{t_m}, x_0, s_{t_m})||q_\theta(x_{t_{m-1}}|x_{t_m}, s_{t_m}))$$

$$- E_{p(S,x_0)} \mathrm{KL}(p(x_1|s_1, x_0)||p(x_\infty)). \qquad (12)$$

*This objective is minimized when $q_\theta(x_{t_{m-1}}|x_{t_m}, s_{t_m}) = p(x_{t_{m-1}}|x_{t_m}, s_{t_m})$.*

*Proof.* Just as with the classical ELBO, we can write

$$E_{p(x_0)} \log q(x_0) \geq E_{p(x_0,S)} E_{p((x_t)_{t\in[0,1]},S|x_0)} \log \frac{q_\theta((x_t)_{t\in[0,1]}, S)}{p((x_t)_{t\in[0,1]}, S|x_0)}. \tag{13}$$

Then we can break it up as in Prop. A.1 to get

$$E_{p(x_0)} \log q(x_0) \geq E_{p((x_t)_t)} \log \frac{q_\theta((x_t)_t|x_1, S)}{p((x_t)_t|x_0, x_1, S)} - \mathrm{KL}(p(S)||q(S))$$

$$E_{p(S|x_0)} \log \frac{p(S)}{p(S|x_0)} - E_{p(S,x_0)} \mathrm{KL}(p(x_1|S, x_0)||q_\theta(x_1|S)). \tag{14}$$

By our definition of the event schedule and $q(S)$, the second and third term on the right are $0$. For the fourth term, clearly $p(x_1|x_0, S) = p(x_1|x_0, s_1)$.

By our definition of $q_\theta$,

$$q_\theta((x_t)_t|x_1, S) = \prod_{m=1}^{M} q(x_{t_{m-1}}|x_{t_m}, s_{t_m}).$$

As in teh proof of Prop. A.3, we can write

$$p((x_t)_t|x_0, x_1, S) = \prod_{m=1}^{M} p(x_{t_{m-1}}|x_0, x_1, S, x_{t_{m:M}}) = \prod_{m=1}^{M} p(x_{t_{m-1}}|x_0, s_{t_m}, x_{t_m})$$

where the last equality follows by the Markov property. Thus the first term is

$$\sum_{m=1}^{M} \log \frac{q(x_{t_{m-1}}|x_{t_m}, s_{t_m})}{p(x_{t_{m-1}}|x_0, s_{t_m}, x_{t_m})} = -\sum_{m=1}^{M} \mathrm{KL}(p(x_{t_{m-1}}|x_0, s_{t_m}, x_{t_m})||q(x_{t_{m-1}}|x_{t_m}, s_{t_m})).$$

$\square$

**Proposition A.5.** *(Proof of Prop B.1) $p(x_t|x_t, x_0, s_t)$ factorizes as $\prod_{d=1}^{D} p(\mathrm{pr}(x_t^d)|x_t^d, x_0^d, s_t^d)$ and, when marginalizing over $x_0$, each dimension of $x_{t_{m-1}}$ is independent:*

$$p(\mathrm{pr}(x_t)|x_t, s_t) = \prod_{d=1}^{D} p(\mathrm{pr}(x_t^d)|x_t, s_t).$$

*Proof.*

$$p(\mathrm{pr}(x_t)|x_t, x_0, s_t) = \frac{p(\mathrm{pr}(x_t^d)|x_0, s_t)p(x_t|\mathrm{pr}(x_t^d))}{p(x_t|x_0, s_t)} = \prod_{d=1}^{D} \frac{p(\mathrm{pr}(x_t^d)|x_0^d, s_t^d)p(x_t^d|\mathrm{pr}(x_t^d))}{p(x_t^d|x_0^d, s_t)}$$

which equals $\prod_{d=1}^{D} p(\mathrm{pr}(x_t^d)|x_t^d, x_0^d, s_t^d)$. The second claim follows from integrating the later expression. $\square$

**Proposition A.6.** *(Proof of Prop. 4.4) Define, if $s_t^d > 0$, $\mathrm{pr}(x_t^d)$ as the state at the last event in dimension $d$. Then the first term of Eqn. 4 is*

$$-E_{t\sim\mathrm{Unif}(0,1)} E_{p(x_t,x_0,S)} \frac{\beta_t}{\int_0^t \beta_s ds} \sum_d s_t^d \mathrm{KL}(p(\mathrm{pr}(x_t^d)|x_t^d, s_t^d, x_0^d)||q_\theta(\mathrm{pr}(x_t^d)|x_t, s_t)). \tag{15}$$

*Proof.* Call $S^d = \{t_1^d, \ldots, t_{M^d}^d\}$. The first term of Eqn. 4 can be written as

$$-E_{p((x_t)_t,S,x_0)} \sum_{d=1}^{D} \sum_{m=1}^{M^d} \mathrm{KL}(\mathrm{pr}(x_{t_m^d}^d)|x_{t_m^d}^d, x_0^d, s_{t_m^d}^d)||q_\theta(\mathrm{pr}(x_{t_m^d}^d)|x_{t_m^d}, s_{t_m^d})).$$

The term in the sum can be written as $L(s_t, x_t, x_0, d)$ so we can write

$$E_{p((x_t)_t,S,x_0)} \sum_{d=1}^{D} \sum_{t\in S^d} L(s_t, x_t, x_0, d) = \sum_{d=1}^{D} E_{p(S^d)} \sum_{t\in S^d} E_{p(x_0)p(S^{-d})p(x_t|x_0,s_t)} L(s_t, x_t, x_0, d).$$

Call the function after $\sum_{t \in S^d}$ equal to $C(t, s_t^d)$ so we can write the loss as $E_{p(S^d)} \sum_{t \in S^d} C(t, s_t^d)$. We now investigate the measure $E_{p(S^d)} \sum_{t \in S^d}$. First note that $E_{p(S^d)} \sum_{t \in S^d}$ is clearly absolutely continuous in $t$ with respect to the Lebesgue measure so this expression can be written as $E_{t \sim \text{Unif}(0,1)} \sum_{s_t^d} f(t, s_t^d) C(t, s_t^d)$ for some function $f$. By the Lebesgue differentiation theorem, almost everywhere,

$$
\begin{aligned}
f(t', s) &= \lim_{\epsilon \to 0} E_{p(S^d)} \sum_{t \in S^d} \mathbb{1}(t \in [t' - \epsilon, t'], s_{t'}^d = s)/\epsilon \\
&= p(s_{t'}^d = s) \lim_{\epsilon \to 0} E \left[\# \text{ events in } [t' - \epsilon, t'] | s_{t'}^d = s \right]/\epsilon.
\end{aligned}
\tag{16}
$$

The distribution of events on an interval $[0, t]$ is a Poisson process with density $\mu(s) = r\beta_s$; we can simulate this by drawing $s_t \sim \text{Pois}(\int_0^t \beta_s ds)$ and then distributing the $s_t^d$ events with probability according to $\mu/\mu([0, t])$. Therefore, conditioned on $s$ events occurring on $[0, t']$, the density of events occurring at $[t' - \epsilon, t']$ is $\mu(t')/\mu([0, t'])$, that is, the expectation in Eqn. 16 is

$$
s \text{ events} \times \frac{\mu(t')}{\mu([0, t'])} \text{ mass} = s \frac{\beta_{t'}}{\int_0^{t'} \beta_s ds}.
$$

Subbing this into the previous equation completes the proof. $\qquad \square$

**Proposition A.7.** *(Proof of Prop. 5.1) Defining $\alpha_t = \exp(-\int_0^t \beta_s ds)$, the objective in Eqn. 6 is*

$$
E_{t \sim \text{Unif}(0,1)} E_{p(m_t)} E_{p(x_t|x_0,m_t)} \frac{\beta_t \alpha_t}{1 - \alpha_t} \sum_d x_0^T \log \tilde{x}_{0,\theta}(x_t, m_t)^d.
$$

*Proof.* If $s_t^d > 1$ then $\text{pr}(x_t^d)$ is corrupted so $p(\text{pr}(x_t^d)|x_t^d, s_t^d, x_0^d)$ is a uniform categorical and doesn't depend on $x_0$; therefore, by our parameterization of $q_\theta$, we have that the KL term in the loss Eqn. 6 is non-zero if and only if $s_t^d = 1$. As well, when $s_t^d = 1$, $p(\text{pr}(x_t^d)|x_t^d, s_t^d = 1, x_0^d) = \delta_{x_0}$. In this case we can write the loss as

$$
E_{t \sim \text{Unif}(0,1)} \frac{\beta_t}{\int_{s<t} \beta_s ds} E_{p(S)} E_{p(x_t|S,x_0)} \sum_d \mathbb{1}(s_t^d = 1) x_0^T \log \tilde{x}_{0,\theta}(x_t, s_t)^d.
$$

Finally note that when $\tilde{x}_{0,\theta}(x_t, s_t)$ predicts $x_0$, $s_t$ is only useful in telling the model which tokens are corrupted. If we call $m_t = s_t > 0$ an indicator of which tokens have been corrupted, then we can parameterize our prediction as $\tilde{x}_{0,\theta}(x_t, m_t)$.

Note $p(x_t|x_0, S) = p(x_t|x_0, m_t)$, so

$$
\begin{aligned}
E_{p(S)} E_{p(x_t|S,x_0)} \sum_d \mathbb{1}(s_t^d = 1) x_0^T \log \tilde{x}_{0,\theta}(x_t, m_t)^d = \\
E_{p(m)} E_{p(x_t|m_t,x_0)} \sum_d p(s_t^d = 1|m_t^d) x_0^T \log \tilde{x}_{0,\theta}(x_t, m_t)^d.
\end{aligned}
\tag{17}
$$

$s_t \sim \text{Pois}(\int_0^t \beta_s ds)$ so $p(s_t^d = 1|m_t^d) = 0$ if $m_t^d = 0$ and

$$
p(s_t^d = 1|m_t^d) = p(s_t^d = 1|s_t^d \geq 1) = \frac{\int_0^t \beta_s ds \, \alpha_t}{1 - \alpha_t}.
$$

$\qquad \square$

**Proposition A.8.** *(Proof of Prop. 5.2) As $\gamma \to 0$ the objective in Eqn. 6 converges to*

$$
-E_{t \sim \text{Unif}(0,1)} E_{p(x_0,x_t)} \beta_t \sum_d \left[ \sum_{b \neq x_t^d} \mathcal{L}_{b,x_t^d} \left( \tilde{s}_{\theta,b}^d - \frac{p(x_t^d = b|x_0^d)}{p(x_t^d|x_0^d)} \log \tilde{s}_{\theta,b}^d - g \left( \frac{p(x_t^d = b|x_0^d)}{p(x_t^d|x_0^d)} \right) \right) \right]
$$

*where $g(x) = x(\log x - 1)$.*

*Proof.* Note $s_t \sim \text{Pois}(r^* \int_0^t \beta_s ds / \gamma)$, so, as $\gamma \to 0$, $s_t \gamma$ converges to $r^* \int_0^t \beta_s ds$.

As $\gamma \to 0$,

$$K^{s_t} = (I + \gamma \mathcal{L}/r^*)^{s_t} = \exp(\gamma s_t \mathcal{L}/r^*) + o(\gamma) \to \exp\left(\int_0^t \beta_s ds \mathcal{L}\right) = Q_t,$$

where $Q_t$ is the matrix where $Q_{t,b,b'} = p(x_t = b'|x_0 = b)$.

$$
\begin{aligned}
q_\theta(\text{pr}(x_t^d)|x_t, s_t) &= \frac{K_\gamma x_t^d \circ K_\gamma^{s_t - 1} \tilde{x}_{0,\theta}}{x_t^{d,T} K_\gamma^{s_t} \tilde{x}_{0,\theta}} \\
&= \frac{K_\gamma x_t^d \circ K_\gamma^{-1} Q_t \tilde{x}_{0,\theta}}{x_t^{d,T} Q_t \tilde{x}_{0,\theta}} + o(\gamma) \\
&= x_t + K x_t^d \circ \tilde{s}_\theta^d - x_t^d \circ K \tilde{s}_\theta^d + o(\gamma) \\
&= x_t + \gamma \left(\mathcal{L} x_t^d \circ \tilde{s}_\theta^d - x_t^d \circ \mathcal{L} \tilde{s}_\theta^d\right) + o(\gamma).
\end{aligned}
\tag{18}
$$

The expression for $p(\text{pr}(x_t^d)|x_t^d, x_0^d, s_t^d)$ is identical replacing $\tilde{x}_{0,\theta}$ with $x_0$. Thus

$$
\begin{aligned}
&-\text{KL}(p(\text{pr}(x_t^d)|x_t^d, s_t^d, x_0^d)||q_\theta(\text{pr}(x_t^d)|x_t, s_t)) \\
&= \sum_{b \neq x_t^d} \gamma \mathcal{L}_{b,x_t^d} \frac{p(x_t^d = b|x_0^d)}{p(x_t^d|x_0^d)} \log \frac{\tilde{s}_{\theta,b}^d}{p(x_t^d = b|x_0^d)/p(x_t^d|x_0^d)} \\
&\quad + (1 - O(\gamma)) \log \frac{1 + \gamma \left(\mathcal{L}_{x_t^d,x_t^d} - x_t^d \mathcal{L} \tilde{s}_\theta^d\right)}{1 + \gamma \left(\mathcal{L}_{x_t^d,x_t^d} - x_t^d \mathcal{L}(p(x_t^d = b|x_0^d)/p(x_t^d|x_0^d))_b\right)} + o(\gamma) \\
&= \gamma \left[\sum_{b \neq x_t^d} \mathcal{L}_{b,x_t^d} \left(\tilde{s}_{\theta,b}^d - \frac{p(x_t^d = b|x_0^d)}{p(x_t^d|x_0^d)} \log \tilde{s}_{\theta,b}^d - g\left(\frac{p(x_t^d = b|x_0^d)}{p(x_t^d|x_0^d)}\right)\right)\right] + o(\gamma).
\end{aligned}
\tag{19}
$$

Multiplying this by $s_t^d$, we get $\gamma s_t^d \to \int_0^t \beta_s ds$. $\qquad\square$

# B DETAILS OF METHOD

Here we describe how we sample and pick $\beta_t$ for SCUD as described in Sec. 4.3.

## B.1 ALGORITHM FOR ESTIMATING ELBO

We calculate $p(x_\infty)$ from an spectral decomposition of $K$, or, if $K$ is very large, using power iteration.

## B.2 PARAMETERIZATION

First we show that $p(\text{pr}(x_t^d)|x_0, s_t, x_t)$ factorizes across its dimensions.

**Proposition B.1.** *(Proof in Prop A.5 in the Appendix)* $p(x_t|x_t, x_0, s_t)$ *factorizes as* $\prod_{d=1}^D p(\text{pr}(x_t^d)|x_t^d, x_0^d, s_t^d)$ *and, when marginalizing over $x_0$, each dimension of $x_{t_{m-1}}$ is independent:*

$$p(\text{pr}(x_t)|x_t, s_t) = \prod_{d=1}^D p(\text{pr}(x_t^d)|x_t, s_t).$$

Recall this allows us to parameterize $q_\theta(\text{pr}(x_t)|x_t, s_t)$ so it also factorizes as $\prod_{d=1}^D q_\theta(\text{pr}(x_t^d)|x_t, s_t)$.

We parameterize $q_\theta(\text{pr}(x_t^d)|x_t, s_t)$ to predict

$$\frac{p(x_0^d|x_t, s_t)}{p(x_t^d|x_0^d, s_t^d)} = \frac{p(x_t^d, s_t^d|x_0^d, x_t^{-d}, s_t^{-d})}{p(x_t^d|x_0^d, s_t^d)p(x_t^d, s_t^d)} p(x_0^d|x_t^{-d}, s_t^{-d}) = \frac{p(x_t^d|x_0^d, x_t^{-d}, s_t)p(s_t^d)}{p(x_t^d|x_0^d, s_t^d)p(x_t^d, s_t^d)} p(x_0^d|x_t^{-d}, s_t^{-d}).$$

---

**Algorithm 1** Unbiased estimate of the SCUD ELBO (Eqn. 4) using Prop. 4.4

---

**Input:** $x_0$
$S \sim p(S)$
$t \sim \text{Unif}(0, 1)$
// Sample $x_t$
**for** $d = 1, \dots, D$ **do**
    $x_t^d \sim \text{Categorical}(K^{s_t^d} x_0^d)$
**end for**
// Denoise one event of each dimension of $x_t$
Predict $\tilde{x}_{0,\theta}(x_t, s_t)$
**for** $d = 1, \dots, D$ **do**
    Calculate $q_\theta(\text{pr}(x_t^d)|x_t^d, s_t^d)$                        $\triangleright$ use Eqn. 7
    Calculate $p(\text{pr}(x_t^d)|x_t^d, s_t^d, x_0^d)$               $\triangleright$ use Eqn. 5
    Calculate $p(x_1^d|s_1^d, x_0^d) = \text{Categorical}(K^{s_1^d} x_0^d)$.
**end for**
**Return:**

$$-\sum_{d=1}^{D} \left( \frac{s_t^d \beta_t}{\int_0^t \beta_s ds} \text{KL}(p(\text{pr}(x_t^d)|x_t^d, s_t^d, x_0^d)||q_\theta(\text{pr}(x_t^d)|x_t^d, s_t^d)) + \text{KL}(p(x_1^d|s_1^d, x_0^d)||p(x_\infty)) \right).$$

---

Now note $p(x_t^d|x_0^d, x_t^{-d}, s_t) = p(x_t^d|x_0^d, s_t)$ and $p(s_t^d)/p(x_t^d|s_t^d)$ does not depend on $x_0^d$. Thus we aim to predict a quantity proportional to $p(x_0^d|x_t^{-d}, s_t^{-d})$; we call our prediction $\tilde{x}_{0,\theta}(x_t, s_t)$, which we plug into Eqn. 7 and then normalize.

### B.3 Sampling

To sample a point $x_0 \sim q(x_0)$ we first sample the noised sample $x_1 \sim q(x_1) = p(x_\infty)$ and the number of events in each dimension $S \sim q(S) = p(S)$. We now sample given a budget of $C$ evaluations of $\tilde{x}_{0,\theta}$. Every step we denoise the last $\lceil s_1/C \rceil$ events that have yet to be denoised. To denoise we can use Eqn. 7. In the case that we denoise $k \geq 1$ events for a dimension $d$ at once, we can use the fact that

$$p(\text{pr}^k(x_t^d)|x_t, s_t) = \sum_{x_0^d} p(\text{pr}^k(x_t^d)|x_t^d, s_t^d, x_0^d) p(x_0^d|x_t, S)$$

$$= \sum_{x_0^d} p(x_t^d|\text{pr}^k(x_t^d)) p(\text{pr}^k(x_t^d)|s_t^d, x_0^d) \frac{p(x_0^d|x_t, s_t)}{p(x_t^d|s_t^d, x_0^d)}.$$

We can write

$$p(x_t^d|\text{pr}^k(x_t^d)) = \text{pr}^k(x_t^d)^T K^k x_t^d$$

$$p(\text{pr}^k(x_t^d)|s_t^d, x_0^d) = x_0^{d,T} K^{s_t^d-k} \text{pr}^k(x_t^d)$$

And we can approximate the fraction with $\tilde{x}_{0,\theta}$ just as in Eqn. 7. Thus we define

$$q_\theta(\text{pr}^k(x_t^d)|x_t, s_t) = K^k x_t^d \circ K^{s_t^d-k,T} \tilde{x}_{0,\theta}. \tag{20}$$

The total procedure is summarized in Alg. 2.

### B.4 Choosing the rate

**Mutual information rate functions** To choose the rate function $\beta_t$, Austin et al. (2021) calculated the frequency of tokens in the training data $p_0(b)$ and then calculated the joint distribution of $x_0$ and a particle which has evolved according to $\mathcal{L}$ for time $\tau$ along one dimension –

$$p(x_0 = b, x_\tau = b') = p_0(b)(e^{\tau \mathcal{L}})_{b,b'}.$$

---

**Algorithm 2** Efficient sampling from SCUD

---

**Input:** function evaluation budget $C$.
// Sample $x_1, s_1$
**for** $d = 1, \ldots, D$ **do**
    $x^d \sim p(x_\infty)$
    $s^d \sim p(s_1) = \text{Pois}(\int_0^1 \beta_s ds)$
**end for**
$L \leftarrow \lceil \sum_{d=1}^D s^d / C \rceil$                  $\triangleright$ Number of events to denoise per step
**for** $c = 1, \ldots, C$ **do**
    // Decide which positions to denoise in this step
    $k \leftarrow \vec{0}$
    **for** $\ell = 1, \ldots, L$ **do**
        **if** $\sum_{d=1}^D (s^d - k^d) > 0$ **then**            $\triangleright$ If there are remaining events to reverse...
            $d \sim \text{Categorical} \left( \frac{s-k}{\sum_{d=1}^D (s^d - k^d)} \right)$    $\triangleright$ ...sample uniformly from remaining events in $s$.
            $k^d \leftarrow k^d + 1$
        **end if**
    **end for**
    // Denoise $k^d$ steps at each dimension $d$
    Predict $\tilde{x}_{0,\theta}(x, s)$
    **for** $d = 1, \ldots, D$ **do**
        $x^d \sim q_\theta(\text{pr}^{k^d}(x^d)|x, s)$               $\triangleright$ use Eqn. 20
        $s^d \leftarrow s^d - k^d$
    **end for**
**end for**
**Return:** $x$

---

They calculate the mutual information function $\text{MI}(\tau)$ of this joint distribution; the mutual information is normalized so $\text{MI}(0) = 1$. They then pick $\beta_t$ so that evolving in the modulated process linearly decreases the mutual information from 1 to $\epsilon$ on the interval $[0, 1]$, i.e. $\text{MI}(\int_0^t \beta_s ds) = 1 - (1 - \epsilon)t$. For clarity, we'll set $\text{MI}(\int_0^t \beta_s ds) = 1 - t$ and look at the interval $[0, 1 - \epsilon]$ below.

**Implementation in continuous time** The process in (Austin et al., 2021) has discrete time, so the integral over $\beta$ is a sum and each $\beta_t$ can be pre-calculated before training begins. When we implement continuous time discrete diffusion, we use a Newton root finder to calculate $\int_0^t \beta_s ds = \text{MI}^{-1}(1-t)$ and the implicit function theorem to calculate $\beta_t = \frac{d}{dt} \int_0^t \beta_s ds = 1/\left( \frac{d}{dt} \text{MI}(\int_0^t \beta_s ds) \right)$.

**Schedules for SCUD** For SCUD, we instead calculate the joint distribution between $x_0$ and the particle after $m$ events, $x_{t_m}$, along one dimension –

$$p(x_0 = b, x_{t_m} = b') = p_0(b)(K^m)_{b,b'}.$$

Calling the mutual information between these variables $\text{MI}_m$ we choose $\beta_t$ so that $E_{s_t} \text{MI}_{s_t} = 1 - t$ where $s_t \sim \text{Pois}(r^* \int_0^t \beta_s ds / \gamma)$. Again we calculate these values using a Newton root finder and the implicit function theorem.

**Connection to classical discrete diffusion** With this choice, note as $\gamma \to 0$, for any $\tau$

$$\text{MI}_{r^* \tau / \gamma} = \text{MI}(p_0(b)((I + \gamma \mathcal{L}/r^*)^{r^* \tau / \gamma})_{b,b'}) \to \text{MI}(p_0(b)(e^{\tau \mathcal{L}})_{b,b'}) = \text{MI}(\tau).$$

Therefore, $E_{s_t} \text{MI}_{s_t} \to \text{MI}(\int_0^t \beta_s ds)$, so $\int_0^t \beta_s ds$ converges to the same value as in classical discrete diffusion.

**Connection to masking discrete diffusion** In this case, $x_{t_m}$ is uniform independent of $x_0$ for all $m \geq 1$ Therefore, $\text{MI}_m = 0$ for all $m \geq 1$ and $E_{s_t} \text{MI}_{s_t} = e^{-\int_0^t \beta_s ds} = \alpha_t$. Therefore, $\alpha_t = 1 - t$.

## C  STRUCTURED PROCESSES

In this section we will describe the structured continuous time Markov processes we used in Sec. 7. Our processes are inspired by those from Austin et al. (2021) and Alamdari et al. (2023); however those works framed the process in discrete time in such a way that they are not related to any continuous time Markov model, requiring us to design new processes. Note also that those works modified their processes to ensure that the transition matrix at every time-point was doubly stochastic; this was so that all transition matrices would have the same stationary distribution – a uniform distribution. In our case, we are free to pick any $\mathcal{L}$ that converges to a stationary distribution, even if it is not uniform.

### C.1  GAUSSIAN PROCESS FOR IMAGES

To include the bias that two pixel values $i \neq j$ are similar if $(i - j)^2$ is small, we set $\mathcal{L}_{i,j} = \exp(-200\frac{(i-j)^2}{B})$ the value 200 was chosen as it gave the best results in small scale experiments. We then set $\mathcal{L}_{i,i} = -\sum_{j \neq i} \mathcal{L}_{i,j}$.

### C.2  NEAREST NEIGHBOUR PROCESS FOR LANGUAGE

Our vocabulary in the language result was approximately 30'000 tokens from the Bert-base-uncased tokenizer (Devlin et al., 2018). It is prohibitively expensive to compute a $30'000 \times 30'000$ matrix $K$ to take matrix vector products during training. Instead, we pick a sparse $K$ built using the embeddings from Devlin et al. (2018); for the most frequent 1000 words (which make up 95% of tokens seen in the data) $i, j$ we computed their similarity as $v_i^T v_j$ where $v_i$ is the normalized embedding of word $i$. For each word we found the 10 nearest neighbours; we noticed restricting to the top 1000 words resulted in nearest enighbours which were much more semantically similar. We next set, for nearest neighbours,

$$\tilde{\mathcal{L}}_{i,j} = \exp(v_i^T v_j / 0.3).$$

We next normalized $\tilde{\mathcal{L}}$ so that the diagonal is 1 – this ensures that every word has an identical transition rate, avoiding the case where a word never transitions because it has no nearby neighbours.

We noticed that it often took a long time for particles to reach a stationary distribution with this process, so we added occasional transitions across the nearest neighbour graph; we called $p$ the normalized frequencies of the top 1000 words in the data and define the uniform transition infinitesimal generator

$$\mathcal{L}_{\text{unif}} = \mathbb{1} \otimes p - I,$$

where $\mathbb{1}$ is the vector of all 1's; this transitions tokens to a random token based on the final token's frequency in the data. We combine our two processes by defining

$$\mathcal{L} = \tilde{\mathcal{L}} + 0.4 \times \mathcal{L}_{\text{unif}}$$

and normalizing so that the smallest value on the diagonal was $-1$. We do not store this matrix explicitly, and only perform matrix operations with sparse matrix products and multiplication with $\mathbb{1}$ or $p$.

For tokens outside of the most frequent 1000, we transition using $\mathcal{L}_{\text{unif}}$.

### C.3  BLOSUM PROCESS FOR PROTEIN

BLOSUM is a matrix that can be describes how often different amino acids are seen in the same position in related protein families (Henikoff & Henikoff, 1992). The $i, j$ entry of the matrix is

$$B_{i,j} = 2 \log \frac{P_{ij}}{P_i P_j}$$

where $P_{ij}$ is the probability of two related proteins having amino acids $i, j$ at the same position, and $P_i$ is the marginal probability. We build a stochastic process to emulate drawing a related protein, so we set

$$K_{i,j} = \exp(B_{i,j}/2) \times P_j = P_{j|i}.$$

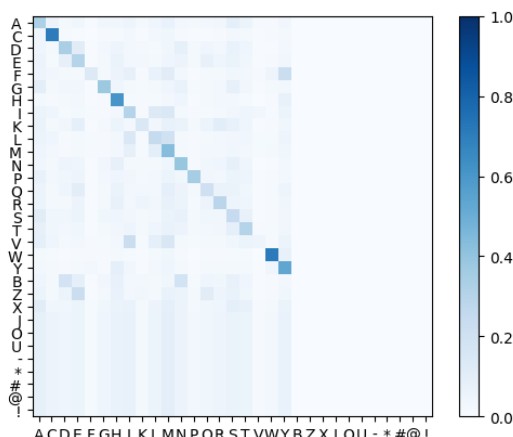

Figure 5: BLOSUM process $K$.

There are other letters in our vocabulary for non-canonical amino acids and padding; for $i$ not one of the canonical 20 amino acids, we set $K_{i,j} = P_j$, so all transitions an only occur to a canonical amino acid. Finally we set $\mathcal{L} = K - I$ (Fig. 5).

# D    EXPERIMENTAL DETAILS

In all cases we trained models on 2 $A$100 GPUs on an academic cluster.

## D.1    TRANSITION RATES IN FIG. 3

For $\tau$LDR we downloaded the CIFAR10 model from https://github.com/andrew-cr/tauLDR. We simulated 2000000 forward trajectories using samples from CIFAR10 and 100 backward samples using the $\tau$ leaping code with 2048 steps. For forward samples $x$ we calculated rates $-\mathcal{L}_{x,x}$ and for backward samples we calculated rates $-\mathcal{L}_{\theta,x,x}$. We then averaged forward and backward rates at each timestep. We finally average with a sliding window of size 9.

For D3PM we download the 640M uniform model from https://github.com/microsoft/evodiff. We were able to calculate the forward rates analytically. As above, we simulated 100 backward samples and at each time step we calculated the probability of a transition; we multiplied this probability by $1/\Delta t$ to get a rate. We averaged as above.

## D.2    IMAGES

We use an architecture inspired by Kingma et al. (2021) like in MD4 (Shi et al., 2024) with a slight modification to incorporate $s_t$. The architecture first embeds $x_0$ like in Shi et al. (2024) and then puts it through a UNet with 32 layers and no up- or down-sampling. At every layer of the UNet, a feed forward layer is applied to a sinusoidal embedding of the time $t$ and the output is added to the channels at every pixel – ax position $i, j$, activations $a^{i,j}$ at updated

$$a^{i,j} \leftarrow \mathrm{FF}_\theta(\mathrm{emb}(t)) + a^{i,j}.$$

Instead each activation is updated using a FiLM layer using the number of events up to time $t$.

$$a^{i,j} \leftarrow \mathrm{FF}_{1,\theta}(\mathrm{emb}(s_t^{i,j})) + \mathrm{FF}_{2,\theta}(\mathrm{emb}(s_t^{i,j})) \circ a^{i,j}.$$

The feed forward layers are shared across every position $i, j$. We used the same training parameters as in Shi et al. (2024); we trained each of our large models for 2 days and each of our models from Fig. 2 took between 1.5 and 2 hours.

We use $K = 2048$ function evaluations to generate images. The results of Fig. 2 used a batch size of 16 and the same architecture but with an 8 layer UNet – masking and classical models used FiLM layers with $t$ instead of $s_t$.

### D.3 LANGUAGE AND PROTEIN

We use the diffusion transformer architecture (Peebles & Xie, 2022) as in SEDD (Luo et al., 2022). This architecture has FiLM layers to add $t$ at each layer; as above, we replace $t$ with $s_t$. We use the training settings as in SEDD (Luo et al., 2022), accumulating to match their batch size of 512. We trained our models for 2 days each.

### D.4 PROTEIN

We use the small CARP architecture from (Alamdari et al., 2023). The original architecture added as embedding of $t$ at the first layer. We add FiLM layers for $s_t$ at every layer as described above. We train and test on the March 2020 release of Uniprot2020 released by Alamdari et al. (2023). We use a batch size of 128 protein up to size 1024 as in Alamdari et al. (2023), randomly truncating proteins over that size. We trained each model for 2 days.

### D.5 COMPUTATIONAL COMPLEXITY

In terms of computational complexity, the major differences between SCUD and classical discrete diffusion are (A) replacing operations of $\mathcal{L}$ with operations of $K$, and (B) replacing the time $t$ in the argument of $\tilde{x}_{0,\theta}$ with the number of transitions $S$. We discuss how (B) does not result in a large increase of computational complexity below, and note that (A) does not change the computational complexity except when the number of tokens $B$ is large, when it actually enables strategies that reduce complexity.

**(A) Matrix computations** To calculate our loss, Eqn. 6, in Eqn. 6 we see that we only need to take matrix vector products with K; the analogous quantity in classical discrete diffusion requires matrix exponentiation $\exp(t\mathcal{L})$ (Luo et al., 2022). When $B$ is small, both these calculations have negligible complexity and can be calculated similarly quickly by precomputing an eigen-decomposition of $K$ or $\mathcal{L}$. But when $B$ is large, as in the language modeling case, these calculations become very expensive; Luo et al., 2022 settled for very simple $\mathcal{L}$, masking and uniform, such that $\exp(t\mathcal{L})$ can be easily analytically calculated; SCUD is able to build in a richer forward process by picking a sparse + low rank $K$ so that matrix vector products are very fast.

In terms of big-O notation, when an eigendecomposition is precomputed, $(\exp(t\mathcal{L})\tilde{x}_0^d)_{d=1}^D$ and $(K^{s_t^d}\tilde{x}_0^d)_{d=1}^D$ each cost $\Theta(DB^2)$ for two dense matrix multiplies and a scaling by the exponentiation or power of the eigenvalues. When $K^{s_t^d}$ is a sparse matrix with $O(rB)$ entries or has a rank of $r$, calculating $(K^{s_t^d}\tilde{x}_0^d)_{d=1}^D$ is $O(DBr\max_d(s_t^d))$; in our language case, $B$ is large while $\max_d(s_t^d) \approx 30$ and we pick $r \approx 20$ resulting in a large speedup.

**(B) Computations with $S$** Indeed, the place that SCUD adds some overhead to calculations is in replacing the arguments of $\tilde{x}_{0,\theta}(x_t, \cdot)$: the time over which $x_0$ has been corrupted, $t$, a scalar, is replaced with the number of corruptions of each token $S$, a $D$-dimensional object. The overhead of this operation is dependent on the architecture of $\tilde{x}_{0,\theta}$. We picked $\tilde{x}_{0,\theta}$ so that no parameters were added by replacing $t$ with $S$, and such that the computational and memory overhead caused by this replacement were negligible compared to the operations and memory spent on operations on the $D$-dimensional $x_t$. Above we used previous architectures modified so that each operation on $t$ was also applied to each dimension of $S$. As well, for the architectures we chose, whenever a function of $t$ was added or multiplied to a set of activations, say at layer $\ell$, $h_{\ell,\theta}$, the activations had a dimension $D$, so we could perform the same operation with element-wise addition or multiplication with $S$, i.e.
$$h_{\ell+1,\theta}^d = f_{1,\theta}(t)h_{\ell,\theta}^d + f_{2,\theta}(t) \text{ was replaced with } h_{\ell+1,\theta}^d = f_{1,\theta}(s_t^d)h_{\ell,\theta}^d + f_{2,\theta}(s_t^d).$$
Thus, adapting $\tilde{x}_{0,\theta}$ for SCUD in this way adds no extra parameters. The overhead of this change is that every call to $f_\theta$ is replaced by $D$ calls, $D$-times the activations $f_\theta(s_t^d)$ must be stored, and $D$-times more gradients must be calculated for $f_\theta(s_t^d)$. $f_\theta$ is however a set of linear functions and activations. The operations on the corrupted data $x_t$ involve convolutions and attention, which have much larger memory and computational costs. In big-O notation, the cost of calculating $\tilde{x}_{0,\theta}(x_t, t)$ and $\tilde{x}_{0,\theta}(x_t, S)$ are therefore identical – at worst, the constant in front of the largest term changes. Therefore, in our experiments, we ran all models for roughly equal time with the same batch sizes and did not observe any substantial difference in computation.

# E    EXTENDING FLOW MATCHING TO SCUD

Here we follow the exposition of Campbell et al. (2024a) to derive flow matching models that are conditioned on schedule. In App. E.1 we derive schedule conditioned flow matching (SCUM) in generality. In App. E.2 we describe how SCUD is an instance of SCUM and show how by training a SCUD model, one can sample from a large class of SCUM models. Finally in App. E.3 we derive an example class of SCUM models. The conclusion is that schedule conditioning can be extended to the flow matching case just as classical discrete diffusion can.

## E.1    SCHEDULE CONDITIONED FLOW MATCHING (SCUM)

We consider discrete objects in a set of size $B$ and in this quick exposition leave out the multi-dimensional case as an easy extension of the logic of SCUD or Campbell et al. (2024a). In flow matching, we wish to approximately sample from a target $p(x_0)$ (this is called $x_1$ in Campbell et al. (2024a)). In regular flow matching, we define distributions of samples noised for time $t$: $p(x_t|x_1)$ (Eqn. 6 of Campbell et al. (2024a)). To condition on the schedule, we instead define distributions of samples that have been noised by $s$ events from $x_1$: $p(x_s|x_0)$. We assume $p(x_s|x_1)$ is close to an easy to sample from distribution $p(x_\infty)$ when $s$ has large entries. In particular, for $s$ with large entries, the marginal $p(x_s) \approx p(x_\infty)$; Now we want to denoise events to get $p(x_{s-1})$ and ultimately $p(x_0)$ (Eqn. 5 of Campbell et al. (2024a)).

To do so, we first choose how to denoise elements in $p(x_s|x_0)$. Say $K_{s|x_0}$ is a stochastic matrix such that sampling $p(x_s|x_0)$ then $x_{s-1} \sim \text{Categorical}(K_{s,d|x_0}^T x_s^d)$ gives a sample from $p(x_{s-1}|x_0)$. The next result is the analogous result of Prop. 3.1 of Campbell et al. (2024a): given a sample from the marginal, $x_s \sim p(x_s)$ we can denoise an event in dimension $d$ by averaging over $x_0|x_s$ and using $K_{s|x_0}$.

**Proposition E.1.** *Define $K_{s;x_s,\cdot} = E_{p(x_0|x_s)}K_{s|x_0;x_s,\cdot}$. Then sampling $x_s \sim p(x_s)$ and $x_{s-1} \sim$* Categorical$(K_{s;x_s,\cdot})$ *gives a sample from $p(x_{s-1})$.*

*Proof.*

$$E_{p(x_s)}E_{p(x|x_s)}K_{s|x_0;x_s,x_{s-1}} = \sum_{x_0} p(x_0)\left(E_{p(x_s|x_0)}K_{s|x_0;x_s,x_{s-1}}\right)$$

$$= \sum_{x_0} p(x_0)p(x_{s-1}|x)$$

$$= p(x_{s-1}).$$

□

Given this result, we can define schedule conditioned flow matching models (SCUM). First we approximate $p(x_0|x_s)$ with a neural network $\tilde{x}_{0,\theta}(x_s, s)$; next we sample from $p(x_\infty)$ which is $\approx p(x_s)$ for some large $s$, and then iteratively denoise by approximating $K_{s;x_s,\cdot}$ (Alg. 3).

---

**Algorithm 3** Sampling from SCUM in analogy to Alg. 1 in Campbell et al. (2024a)

$s \leftarrow$ large number
$x_s \sim p(x_\infty) \approx p(x_s)$
**while** $s > 0$ **do**
    $K_{s;x_s,\cdot} \leftarrow E_{\tilde{x}_{0,\theta}(x_s,s)}K_{s|x_0;x_s,\cdot}$
    $x_{s-1} \sim \text{Categorical}(K_{s;x_s,\cdot})$
    $s \leftarrow s - 1$
**end while**
**Return:** $x_0$

---

To train $\tilde{x}_{0,\theta}(x_s, s)$ we can just minimize the cross entropy

$$E_{s\sim\text{Unif}(1,2,...,\text{large number}),p(x_0),p(x_s|x_0)}x_0^T \log \tilde{x}_{0,\theta}(x_s, s).$$

We could alternatively use a different distribution for $s$, such as a Poisson. Note that $p(x_s|x_0)$ does not depend on the particular choice of $K_{s|x_0}$, so we can train $\tilde{x}_{0,\theta}(x_s, s)$ once and then decide the best $K_{s|x_0}$ for sampling at test time.

### E.2  SCUD IS SCUM

We now show that for a particular choice of $K_{s|x_0}$, the simulated trajectories of SCUM are that of SCUD as in Appendix H of Campbell et al. (2024a). Next we discuss how, given a trained SCUD model we can sample from a wide variety of SCUM models.

Define a Markov process that noises datapoints $x_0$ with an infinitesimal generator $\mathcal{L}$ with rate function $\beta_t$. Say we have a data point $x_0$ that's been noised $s > 0$ times and define $K_{s|x_0;x_s,\cdot} = p(\mathrm{pr}(x_s)|x_s, x_0, s)$ as in Eqn. 5. Then

$$
\begin{aligned}
K_{s;x_s,\cdot} &= E_{p(x_0|x_s)} K_{s|x_0;x_s,\cdot} \\
&= E_{p(x_0|x_s,s)} p(\mathrm{pr}(x_s)|x_s, x_0, s) \\
&= p(\mathrm{pr}(x_s)|x_s, s)
\end{aligned}
$$

which is exactly the distribution we approximate to denoise an event in SCUD (Alg. 2). Therefore SCUD is just SCUM with a particular choice of $K_{s|x_0}$, with "large number" in Alg. 3 set to $\mathrm{Pois}(\int_0^t \beta_s ds)$.

Furthermore, SCUD trains a $\tilde{x}_{0,\theta}(x_s, s)$ to predict $x_0$ given $x_s, s$[3]. Campbell et al. (2024a) suggests that an advantage of flow matching is that one can train $\tilde{x}_{0,\theta}$ once and then decide on the best infinitesimal generator at test time; we can do the same by training $\tilde{x}_{0,\theta}$ with the SCUD objective and then changing $K_{s|x_0}$ at test time.

### E.3  EXAMPLES OF SCUM

Say we have built a SCUD model with transition matrix $K$. The canonical choice for $K_{s|x_0}$ above is

$$
K_{s|x_0;x_s,x_{s-1}} = x_{s-1}^T K x_s \frac{x_0^T K^{s-1} x_{s-1}}{x_0^T K^s x_s}.
$$

We now describe a family of $K_{s|x_0}$ that can be alternatively used to sample from $p(x_0)$.

First note that for SCUD, $p(x_s|x_0) = K^{s,T} x_0$. Therefore $K_{s|x_0}$ can be any matrix with $K_{x|x_0}^T K^{s,T} x_0 = K^{s-1,T} x_0$ and positive entries with rows that add to 1. Campbell et al. (2024a) suggested picking the process to minimally move mass from position with too much in $K^{s-1,T} x_0$ to those with too little in $K^{s,T} x_0$ ($R^*$ in Prop. 3.2 in Campbell et al. (2024a)); we can do that with the choice

$$
K^*_{s|x_0;x_s,y} = \frac{\mathrm{ReLU}(y^T K^{s-1,T} x_0 - y^T K^{s,T} x_0)}{\sum_z \mathrm{ReLU}(z^T K^{s-1,T} x_0 - z^T K^{s,T} x_0)} \times \mathrm{ReLU}(x_s^T K^{s,T} x_0 - x_s^T K^{s-1,T} x_0)
$$

for $x_s \neq y$, which moves mass from $x_s$ with too much mass to $y$ with too little in proportion to how much mass they need.

To augment this "most efficient" choice Campbell et al. (2024a) describe a method to add stochasticity to $K_{s|x_0}$. They do so by introducing an infinitesimal generator that obeys details balance; we do the same. Say $\mathcal{L}^{\mathrm{DB}}_{s|x_0}$ keeps the distribution $p(x_s|x_0)$ stationary, say by satisfying detailed balance. Then we can add more noise to $K_{s|x_0;x_s,y}$ by defining $K^\eta_{s|x_0} = e^{\eta \mathcal{L}^{\mathrm{DB}}} K^*_{s|x_0}$ since

$$
K_{x|x_0}^T K^{s,T} x_0 = K^*_{s|x_0} e^{\eta \mathcal{L}^{\mathrm{DB}}} K^{s,T} x_0 = K^*_{s|x_0} K^{s,T} x_0 = K^{s-1,T} x_0.
$$

By varying $\eta$, Campbell et al. (2024a) optimized samples for stochasticity against likelihood.

---

[3]In the high dimensional case, unlike our exposition of SCUM, SCUD trains $\tilde{x}_{0,\theta}(x_s, s)$ to approximate, for each dimension $d$, $p(x_0^d|x_s^{-d}, s)$ rather than $p(x_0^d|x_s, s)$ (Sec. 4.3). However any prediction of $p(x_0^d|x_s^{-d}, s)$ can be transformed into a prediction of $p(x_0^d|x_s, s)$ via the identity $p(x_0^d|x_s, s) \propto p(x_s^d|s^d, x_0^d) p(x_0^d|x_s, s)$ which doesn't depend on the specific choice of $K_{s|x_0}$ – the difference is just a matter of parameterization.

In conclusion, just as one can do with classical discrete diffusion models, after training a SCUD model, one can optimize a stochasticity parameter $\eta$ to get desirable samples.

