# OpenReview forum: "Improving Discrete Diffusion with Schedule-Conditioning"
_ICLR.cc/2025/Conference — Submitted to ICLR 2025_

### Official Review · Reviewer_ifSU · 2024-10-23

**Soundness:** 2
**Presentation:** 2
**Contribution:** 2
**Rating:** 6
**Confidence:** 4

**Summary:**

This paper attempts to understand the empirical finding that discrete diffusion models using a masking corruption process outperform other noising processes such as uniform or Gaussian noising processes in which tokens are "mutated." The authors hypothesize that masking diffusion dominates due to having knowledge of when corruptions occur, which is not the case for the non-masking noise processes. With this hypothesis, the authors propose a modification to standard non-masking discrete diffusion models that allows the model to condition on information about the schedule of the noising process. The authors derive a new version of the ELBO used to train discrete diffusion models into one that explicitly marginalizes over different noise schedules. This allows them to propose a new training objective and parameterization of the denoising process that conditions on noise schedules. The authors evaluate their approach on images, text, and proteins and demonstrate that their approach consistently improves over models using the same noising process, but without schedule conditioning.

**Strengths:**

The paper is tackling an original problem, namely understanding why masking discrete diffusion outperforms other noising processes including those that have domain-specific inductive biases. While I have some concerns with the hypothesis (see section Weaknesses), I agree that conditioning on noise schedules should help mitigate some of the advantages that masking diffusion models have. The paper contains a rigorous derivation that shows how to incorporate the noise schedules into the modeling framework. The empirical results demonstrate that conditioning on the noise schedule tends to improve likelihood-based metrics over models that use the same noise process but do not incorporate the noise schedule.

**Weaknesses:**

I am not convinced by the primary hypothesis outlined in the abstract that says "...we propose that masking diffusion dominates due to knowledge of when corruptions occur." Unless I am misunderstanding something, masking diffusion models have no knowledge of when an event occurs, only that an event occurred. To put a different way, the identity of the state (masked or not) tells us whether it has been noised. However, we have no idea when the noising occurred. This seems like an important point for motivating the use of noise schedules as input.

The primary weakness for me was the experimental results. In particular, the paper seems to be lacking several key details about the experiments. I will reiterate this in the "Questions" section, but I did not see any explanation for why the authors chose to model Cifar10 with both 128 and 256 pixel values. Based on the lack of explanation, I am left feeling that the B=128 experiments were done simply because this was a setting where the authors found they can get Gaussian SCUD to improve over Masking. However, to me, this makes the results appear cherry-picked. Furthermore, in Figure 2, it appears Gaussian is already outperforming Masking, without SCUD (although perhaps not with statistical significance), diminishing the novelty of the result. There should be some commentary on this since.

All of the experiments appear to be done with different training data than the baselines. Thus, I am not sure any of the numbers are truly valid in terms of comparing between methods. Most concerningly, I didn't see any explanation of this in the text or appendix. Baselines should be redone using the same training data.

For the baselines, it seems a crucial baseline would be to use all of the same hyperparameters as SCUD, but without conditioning. This would use the same noising process (e.g. Gaussian or Uniform), the same architecture, and the same training data but without using any conditioning information (e.g. just pass in a constant $s$ every time).

The authors do not describe how BPD/Perplexity are computed. Since there appears to be an additional variable to marginalize out (the noise schedule), these details are important and need to be explained and justified.

Practically speaking, the field seems to be moving away from discrete diffusion models and more towards discrete flow matching. The latter has a much simpler objective function and training procedure while improving results. However, I am very sympathetic to the fact that field is moving so quickly and therefore do not penalize the authors for this. However, any more discussion about how this can be extended to flow-matching would be welcome and it seems like a straightforward extension?

The writing and presentation could be improved for more clarity. There were a few instances where the writing was too imprecise for me to understand what was meant. For example, around lines 235-236, the authors write "...define pr(x_t^d) as the last event in dimension d...". It was not clear to me what is meant by and "event" and what "last" is referring to. I think what is meant is pr(x_t^d) is the state of x_t before the last noise event?

Another place where the writing could have been more clear is in the "Efficient Loss" section when the authors say "... and then add a weight representing how likely an event is to occur at the instant t." Here I think it would be helpful to write what the term is as I was left confused about which terms in Equation 5 were the weight. I believe the weight term is both the term involving Betas and the multiplication of s_t. As a reader, I was expecting this to be more clearly laid out for me.

**Questions:**

1) Can the authors clarify my point about masking diffusion models not having knowledge of when corruptions occur?

2) Why were experiments done with B=128 and B=256? Was is due to the fact that only for B=128 did SCUD outperform masking?

3) Why were different training data used in the baselines?

4) How were the metrics computed? Specifically, how did the authors handle marginalizing the noise schedule?

---

> ### Author Response · Authors · 2024-11-24
> **Comment 1**
>
> Thank you for your thorough and thoughtful review and suggestions! In our new draft, we incorporated your suggestions for clearer exposition, more experimental baselines, and connecting our method to flow matching. We answer your four questions while describing how we incorporated your suggestions below.
>
>
> >(W1) I am not convinced by the primary hypothesis outlined in the abstract that says "...we propose that masking diffusion dominates due to knowledge of when corruptions occur." Unless I am misunderstanding something, masking diffusion models have no knowledge of when an event occurs, only that an event occurred. To put a different way, the identity of the state (masked or not) tells us whether it has been noised. However, we have no idea when the noising occurred. This seems like an important point for motivating the use of noise schedules as input.
>
>
> >(Q1) Can the authors clarify my point about masking diffusion models not having knowledge of when corruptions occur?
>
>
> Indeed, in Eqn. 2, the schedule conditioned model is shown to predict $\mathrm{prev}(x^d)$ using $S$, the set of transition times. Masking diffusion only sees whether a token has been corrupted but not the full set of times $S$, so how can it be a scheduled conditioned model?
>
>
> Note in Eqn 4, the schedule conditioned model actually only makes its predictions using $s\_t$ the number, but not the times, of transitions up to time $t$. We only pass the number of transitions as this is a sufficient statistic for $S$ – the transition times do not help us better predict the previous token; this is proven in Prop. 4.2 . This is also the case for masking diffusion, which also doesn’t need to take in the exact transition times. Notice we proved this formally in Sec. 5.1. Here we further show that an indicator of whether $s\_t$ is larger than $0$ is a sufficient statistic – this sufficient statistic is the mask.
>
>
> Another way to convince oneself that masking diffusion has baked in the known distribution of transitions is by noting that if we were to recreate the plots in Fig. 3 for a masking diffusion model, we would see no error. During the backwards process in masking, tokens are denoised according to their probability of being noised in the forward process regardless of the prediction $\tilde x\_0$; therefore the noising rate forward and backward are identical. This is not true for uniform for example, where errors in predicting $\tilde x\_0$ can speed up or slow down transitions in the backwards process.
>
>
> >(W2) The primary weakness for me was the experimental results. In particular, the paper seems to be lacking several key details about the experiments. I will reiterate this in the "Questions" section, but I did not see any explanation for why the authors chose to model Cifar10 with both 128 and 256 pixel values. Based on the lack of explanation, I am left feeling that the B=128 experiments were done simply because this was a setting where the authors found they can get Gaussian SCUD to improve over Masking. However, to me, this makes the results appear cherry-picked. Furthermore, in Figure 2, it appears Gaussian is already outperforming Masking, without SCUD (although perhaps not with statistical significance), diminishing the novelty of the result. There should be some commentary on this since.
>
>
> >(Q2) Why were experiments done with B=128 and B=256? Was is due to the fact that only for B=128 did SCUD outperform masking?
>
>
> For the CIFAR10 experiments, we used 128 only because we had to run a large number of experiments and replicates for Fig. 2 and we were limited by compute. We didn’t anticipate results to change substantially at full scale.
>
>
> To confirm this intuition, in the new draft, we replicate the result with 256 in table 1 (we summarize the result in the [global response](https://openreview.net/forum?id=wQk6yaRGOi&noteId=4fknjBKsjZ)). Note 1) with B=128 Gaussian classical diffusion performs similarly to masking; with B=256 however masking diffusion outperforms Gaussian classical diffusion; and 2) in this case, as with B=128, SCUD beats both masking and diffusion.

---

> > ### Author Response · Authors · 2024-11-24
> > **Comment 2**
> >
> > >(W3) All of the experiments appear to be done with different training data than the baselines. Thus, I am not sure any of the numbers are truly valid in terms of comparing between methods. Most concerningly, I didn't see any explanation of this in the text or appendix. Baselines should be redone using the same training data.
> >
> >
> > >(Q3) Why were different training data used in the baselines?
> >
> >
> > For all experiments, we use the exact same training data as the baselines, including the same train / test split and batch sizes. The difference is only that we trained for fewer epochs due to our resource constraints – some of these methods were run for months on large numbers of GPUs. This puts our methods at a disadvantage given that these large baseline models have not been reported to overfit.
> >
> >
> > To confirm this, in our new draft, we ran our image SCUD models for much longer; in the results in the [global response](https://openreview.net/forum?id=wQk6yaRGOi&noteId=4fknjBKsjZ) we see indeed that our methods have much closer perplexities to the previous baseline methods. Furthermore,for images and proteins we included baselines that are trained the same length of time as SCUD and see that SCUD outperforms them.
> >
> >
> > >(W4) For the baselines, it seems a crucial baseline would be to use all of the same hyperparameters as SCUD, but without conditioning. This would use the same noising process (e.g. Gaussian or Uniform), the same architecture, and the same training data but without using any conditioning information (e.g. just pass in a constant  every time).
> >
> >
> > Indeed, our claim is that incorporating knowledge of the noising schedule should improve the fit of the model compared to models without schedule conditioning. Therefore we should compare SCUD to such models. Models that do not take in the noising schedule as an argument are exactly classical discrete diffusion models!
> >
> >
> > Indeed in Fig. 2 we performed this experiment – the Gaussian classical discrete diffusion model had the same training data, architecture, and forward process, only differing in its value of $\gamma=0$ (no conditioning on the schedule) – and we saw that conditioning on the schedule improved the fit. One can also see that we outperformed all previous classical discrete diffusion methods except masking (which does condition on the schedule) in Sec. 7.
> >
> >
> > To strengthen these results, we repeated the experiment in Fig. 2, training our reimplementation of classical discrete diffusion, in the full scale image and protein settings in Sec. 7. For text, as explained in the paper, we could not build a classical diffusion model given that it requires the evaluation of $e^{t\mathcal L}$; for the very large $\mathcal L$ we consider, this calculation would substantially slow down training and we could not train the model in a reasonable time. In the [global response](https://openreview.net/forum?id=wQk6yaRGOi&noteId=4fknjBKsjZ), for image and protein data, we see that given the same data, architecture, and forward process, our SCUD models outperform classical diffusion.
> >
> >
> > >(W7) The writing and presentation could be improved for more clarity. There were a few instances where the writing was too imprecise for me to understand what was meant. For example, around lines 235-236, the authors write "...define pr(x_t^d) as the last event in dimension d...". It was not clear to me what is meant by and "event" and what "last" is referring to. I think what is meant is pr(x_t^d) is the state of x_t before the last noise event?
> >
> >
> > >(W8) Another place where the writing could have been more clear is in the "Efficient Loss" section when the authors say "... and then add a weight representing how likely an event is to occur at the instant t." Here I think it would be helpful to write what the term is as I was left confused about which terms in Equation 5 were the weight. I believe the weight term is both the term involving Betas and the multiplication of s_t. As a reader, I was expecting this to be more clearly laid out for me.
> >
> >
> > Thank you for your suggestions! We’ve rewritten these lines to add a formal definition of  $\mathrm{pr}(x\_t^d)$ and state what we meant by “weight”, which you interpreted correctly, clearly.

---

> > > ### Author Response · Authors · 2024-11-24
> > > **Comment 3**
> > >
> > > >(W5) The authors do not describe how BPD/Perplexity are computed. Since there appears to be an additional variable to marginalize out (the noise schedule), these details are important and need to be explained and justified.
> > >
> > >
> > > >(Q4) How were the metrics computed? Specifically, how did the authors handle marginalizing the noise schedule?
> > >
> > >
> > > In classical diffusion (continuous or discrete), the likelihood is an integral over $t$ which cannot be evaluated analytically. For these models, the integral is empirically estimated by sampling a $t$ for each test sample $x\_0$; given the large size of the test set, this is usually a negligible source of variance. In our case, we take the same approach as previous diffusion models, replacing $t$ with $s\_t$. In our new draft we add this description as an explicit algorithm in App. B.1.
> > >
> > > >(W6) Practically speaking, the field seems to be moving away from discrete diffusion models and more towards discrete flow matching. The latter has a much simpler objective function and training procedure while improving results. However, I am very sympathetic to the fact that field is moving so quickly and therefore do not penalize the authors for this. However, any more discussion about how this can be extended to flow-matching would be welcome and it seems like a straightforward extension?
> > >
> > >
> > > Indeed, a number of exciting new papers have recently built flow matching methods for discrete data (Campbell et al., 2024, Gat et al., 2024)! However, as described below, it’s not yet clear when and if they are superior to discrete diffusion models; rather discrete flow matching may be seen as a set of training and sampling strategies to improve the sample quality of diffusion models. In App E we show the same strategies can be analogously applied to SCUD.
> > >
> > >
> > > The primary stated benefit of discrete flow matching methods is the large design space. The simple objective function and training procedure (straightforwardly predicting $x\_0$ from $x\_t$) for flow models can in fact be used to build diffusion models: this simple procedure is how many diffusion models are built for continuous data (“Simplified training objective“ in Ho et al, 2020), and it also an objective considered for training D3PM (Austin et al., 2021). Newer implementations of masking diffusion, which have not yet been compared with flow matching, also stress their simple training objective (Shi et al, 2024 and Sahoo et al., 2024, also noted in Appendix C of Campbell et al., 2024). The result of using this objective in the continuous case is worse likelihoods but improved sample quality; discrete flow models are evaluated on only sample quality, so their using objective may explain their superior performance to diffusion models. Finally, note that discrete flow matching models have stochastic paths unlike continuous flow models; these may be fundamentally different classes of models so the benefits of continuous flow matching models may not necessarily extend to the discrete case. Therefore, while the development of these flow models are exciting, there is yet to be a systematic comparison of discrete diffusion and flow matching methods!
> > > Indeed, we ran a test and found that the empirical variance of the test likelihood is substantially smaller than the gap between models – for our gaussian SCUD model on images, the variance of our estimate of the BPD on the test set from resampling $S$ for each $x\_0$ was 0.0064.
> > >
> > > *Answer continues in the next comment ...*

---

> > > > ### Author Response · Authors · 2024-11-24
> > > > **Comment 4**
> > > >
> > > > Appendix H of Campbell et al., 2024 investigates the connection between discrete diffusion and discrete flow matching. In their setup, they pick distributions $p(x\_t|x\_0, x\_1)$; pick infinitesimal generators $\mathcal L_{t |x_1}$ that generate these distributions; then evolve $x\_0$ to the data distribution of $x\_1$ (note the reversed roles of $x\_0$ and $x\_1$ compared to diffusion) by evolving with respect to the average $\mathcal L\_{t, x\_t, \cdot}=E\_{p( x\_1|x\_t)}\mathcal L\_{t, | x\_1; x\_t, \cdot}$ by approximating $p(x\_1|x\_t)$. They show that for the distributions $p(x\_t|x\_0, x\_1)$ they consider, a specific choice of $\mathcal L\_{t |x\_0, x\_1}$ can give one discrete diffusion. In discrete flow matching, one may pick any $\mathcal L\_{t |x\_1}$ that doesn’t change $p(x\_1|x\_t)$ to generate samples; in $\tau$LDR diffusion on the other hand, one must directly learn $\mathcal L\_{t, x\_t, \cdot}$ at train time and can’t switch to a potentially better process at test time. More recent discrete diffusion models however, such as SEDD, MD4, and MDLM, all learn a function that predicts $p(x\_1|x\_t)$ and are unaffected by the particular choice of the process $\mathcal L$ used to generate these samples. Therefore, in principle, one could train any of these models to get $\tilde x\_{0, \theta}$ and then switch out $\mathcal L\_{t |x\_0, x\_1}$ to get high quality samples at test time. Therefore flow matching may be seen as a set of training and sampling strategies to improve the sample quality of diffusion models.
> > > >
> > > >
> > > > In a new App E we show we can apply the same techniques to SCUD to build schedule conditioned flow matching (SCUM). Therefore, the same flow matching and training techniques the field is actively exploring to enhance the sample quality of diffusion models can also be used to improve the samples of SCUD. As the field explores the design space of discrete flow matching, just as SCUD gave us insights into discrete diffusion, SCUM can offer another lens to better understand flow matching.
> > > >
> > > >
> > > > **Conclusion**
> > > >
> > > >
> > > > We hope our clarifications and additional experiments were helpful and we hope that you will consider raising your score in light of these additions. If you have any remaining questions or concerns, please raise them so that we can offer further clarifications.
> > > >
> > > >
> > > > **Citations**
> > > >
> > > >
> > > > Ho, Jonathan, Ajay Jain, and Pieter Abbeel. 2020. “Denoising Diffusion Probabilistic Models.” In 34th Conference on Neural Information Processing Systems.
> > > >
> > > >
> > > > Austin, Jacob, Daniel D. Johnson, Jonathan Ho, Daniel Tarlow, and Rianne Van Den Berg. 2021. “Structured Denoising Diffusion Models in Discrete State-Spaces.” Advances in Neural Information Processing Systems 34: 17981–93.
> > > >
> > > >
> > > > Campbell, Andrew, Jason Yim, Regina Barzilay, Tom Rainforth, and Tommi Jaakkola. 2024. “Generative Flows on Discrete State-Spaces: Enabling Multimodal Flows with Applications to Protein Co-Design.” In Proceedings of the 41st International Conference on Machine Learning. arXiv. https://doi.org/10.48550/ARXIV.2402.04997.
> > > >
> > > >
> > > > Gat, Itai, Tal Remez, Neta Shaul, Felix Kreuk, Ricky T. Q. Chen, Gabriel Synnaeve, Yossi Adi, and Yaron Lipman. 2024. “Discrete Flow Matching.” In 38th Conference on Neural Information Processing Systems.
> > > >
> > > >
> > > > Shi, Jiaxin, Kehang Han, Zhe Wang, Arnaud Doucet, and Michalis K. Titsias. 2024. “Simplified and Generalized Masked Diffusion for Discrete Data.” arXiv [Cs.LG]. arXiv. http://arxiv.org/abs/2406.04329.
> > > >
> > > > Sahoo, Subham Sekhar, Marianne Arriola, Yair Schiff, Aaron Gokaslan, Edgar Marroquin, Justin T. Chiu, Alexander Rush, and Volodymyr Kuleshov. 2024. “Simple and Effective Masked Diffusion Language Models.” arXiv [Cs.CL]. arXiv. http://arxiv.org/abs/2406.07524.

---

> > > > > ### Comment · Reviewer_ifSU · 2024-11-26
> > > > >
> > > > > Thank you for the clarification and the additional experiments. I think this paper is addressing a quite interesting problem and the with the additional clarifications about the noise schedule, I do believe that the method is sound and insightful.
> > > > >
> > > > > My remaining concern is still the experiments. As I mentioned before, comparing between models with different training sets is not very insightful. In light of this, the authors have added additional attempts to compare between SCUD and non-SCUD models with the same training set. However, with the smaller training set the behavior is quite different than on the larger training set. For example, in the protein example, BLOSUM now outperforms Masking on smaller training data despite the fact that at larger training set sizes Masking performs best. Thus, it's a bit unclear if the trend of SCUD outperforming non-SCUD BLOSUM will continue. The authors have offered to keep training the models through the review period which may add more insight. Nonetheless, the image results are encouraging and show a clear improvement between non-SCUD and SCUD.
> > > > >
> > > > > Overall, I am encouraged by the changes the authors made and the more apples-to-apples baseline comparisons. I will raise my score.

---

### Official Review · Reviewer_3fup · 2024-11-03

**Soundness:** 3
**Presentation:** 2
**Contribution:** 2
**Rating:** 6
**Confidence:** 1

**Summary:**

This paper propose that masking diffusion succeeds because it leverages information about when corruption events occurred, enabling it to focus on modeling relationships between tokens rather than inferring corruption events. Building on this insight, they introduce a new method called schedule-conditioned diffusion (SCUD), which incorporates corruption schedule information into discrete diffusion models. Experimental results demonstrate that SCUD generalizes masking diffusion and outperforms classical discrete diffusion methods on various datasets.

**Strengths:**

​1. **Innovative Insight**: This paper provides a novel explanation for the success of masking diffusion, linking it to the modeling of corruption schedules, which is a valuable addition to the understanding of discrete diffusion processes.

​2. **Methodology**: The introduction of SCUD is rigorous, with mathematically supported schedule conditioning, which extends masking diffusion and further enhances its performance.

​3. **Empirical Evidence**: Experiments on image, text, and protein data show that SCUD outperforms standard discrete diffusion models, supporting its claims of enhanced generative capability.

**Weaknesses:**

​1. **Lack of Motivation Explanation**: The authors seem to focus heavily on explaining how SCUD works, with less emphasis on why this approach is chosen and what its ultimate goal is. This may make it difficult for readers to follow the authors’ line of reasoning.

​2. **Over-Reliance on Appendix for Proofs**: Many key proofs and details are placed in the appendix, which disrupts the main text’s coherence and could interfere with readers’ logical understanding.

​3. **Inadequate Training/Sampling Procedure Description**: The description of the training and sampling processes is not sufficiently detailed, making it difficult to understand SCUD’s training and sampling mechanics without referring to supplementary materials or  external resources.

**Questions:**

​1.	Could the authors clarify the role of $\Delta t$ in discrete Markov process? This is not clearly defined in the “Background” section, nor are there references provided for further reading.

​2.	Would it be possible to simplify some of the proofs to make them more accessible in the main text?

​3.	Why didn't provide the complete training and inference process in algorithmic form in the paper?

---

> ### Author Response · Authors · 2024-11-24
> **Comment**
>
> Thank you for your thoughtful review! In our new draft, we’ve incorporated your suggestions to make the paper more easily readable. We describe how we’ve incorporated each of your suggestions below.
>
>
> >(Q1) Could the authors clarify the role of delta t in discrete Markov process? This is not clearly defined in the “Background” section, nor are there references provided for further reading.
>
>
> Every time-homogenous Markov process in discrete space can be simulated by what’s known as the “Gillespe algorithm”. This algorithm chooses when to jump and where to jump by sampling exponential random variables. Simply put:
>
>
>     While true:
>         For each b ≠ x_t:
>             Sample Δt_b ~ Exp(L_{x_t, b})
>         Δt = min_b(Δt_b)
>         x_{t+Δt} = argmin_b(Δt_b)
>         t = t + Δt
> We’ve made the description of the process clearer and added a citation to the Gillespe algorithm to Sec. 2.
>
>
> >​(W1) Lack of Motivation Explanation: The authors seem to focus heavily on explaining how SCUD works, with less emphasis on why this approach is chosen and what its ultimate goal is. This may make it difficult for readers to follow the authors’ line of reasoning.
>
>
> To make this roadmap and motivation more clear, we’ve added a paraphrase of the outline below at the end of the introduction, the last paragraphs of Sec. 3, and the first paragraph of Sec 4.
>
>
> In the current paper, Sec. 3 introduces the issue we are trying to solve – discrete diffusion models cannot easily learn the transition schedule – and we quantify the problem in Eqn. 2. We note Eqn 2 suggests a method that solves this problem and goes on to develop this method – SCUD – rigorously in Sec. 4. The main challenge we want to solve is the intractability of $p((x\_t)|x\_1, S)$ and Sec. 4.1 solves this by introducing the concept of events rather than transitions. Sec. 5-7 are dedicated to comparing SCUD to previous methods.
>
>
> >(W2) Over-Reliance on Appendix for Proofs: Many key proofs and details are placed in the appendix, which disrupts the main text’s coherence and could interfere with readers’ logical understanding.
>
>
> >(Q2) Would it be possible to simplify some of the proofs to make them more accessible in the main text?
>
>
> We appreciate your suggestion to better integrate our results in the main text.
>
>
> Of the proofs and details we’ve left to the appendix all either describe minor modifications of standard techniques described elsewhere which we’ve only included for completeness (details in Sec. 4.3), are lengthy but standard algebra (Prop. 3.1, 4.2, 4.3), or require the introduction of formal mathematical techniques and definitions not necessary for understanding the main ideas of the paper (Prop 4.1, 4.4, 5.1, 5.2). However, we’ve added hyperlinks from every result to its proof and from the proof back to the result, and we’ve added an intuitive description of the proof of every result in the paragraph preceding.
>
>
> >​(W3) Inadequate Training/Sampling Procedure Description: The description of the training and sampling processes is not sufficiently detailed, making it difficult to understand SCUD’s training and sampling mechanics without referring to supplementary materials or external resources.
>
>
> >(Q3) Why didn't provide the complete training and inference process in algorithmic form in the paper?
>
>
> While details to replicate our training and sampling are available in the text and code of the paper, we’ve made these clearer so that a reader skimming the paper can immediately identify how to train and sample. We’ve also included both sampling and loss calculations as algorithms.
>
>
> In the current paper, section 4.3 describes how to sample; As described in this section, while the procedure is simple, it uses some extra notation that, while standard (for example, see Campbell et al., 2022 and Zhao et al. 2024), are not necessary for understanding the paper; we’ve put these details in App B.2 for the interested reader. To App. B.3, by your suggestion, we’ve also added the sampling procedure in algorithmic form.
>
>
> For inference, as described in Sec. 4.2 under the section “Efficient loss”, we perform minibatch gradient descent on the loss in Eqn. 6; we evaluate this quantity using the identities in Sec. 4.3. By your suggestion, we’ve highlighted Eqn. 6 as the loss we are optimizing and added the procedure to calculate the ELBO in algorithmic form in App B.1.
>
>
> **Conclusion**
>
> We hope you will look at our global response and updated draft, which contain new experimental results, and we ask that you consider raising your score if your major concerns have been addressed. If you have any remaining questions, we hope you will ask them so that we can offer clarifications.

---

> > ### Comment · Reviewer_3fup · 2024-11-25
> >
> > I like your response, but due to my lack of background knowledge and limited time, I am not able to evaluate it well. I have increased the score and further lowered the confidence, hoping that ACs will pay more attention to the responses from other reviewers.

---

### Official Review · Reviewer_bGrB · 2024-11-04

**Soundness:** 3
**Presentation:** 2
**Contribution:** 3
**Rating:** 6
**Confidence:** 2

**Summary:**

The paper presents Schedule Conditioned Diffusion (SCUD), a new method to enhance discrete diffusion models for conditional sequence generation. The authors identify that existing structured and state-dependent approaches are often outperformed by the simpler "masking diffusion," due to its effective alignment of forward and backward transition schedules. They introduce a decomposition of the Evidence Lower Bound (ELBO) that addresses this mismatch and demonstrate efficient training strategies for SCUD. The findings indicate that SCUD achieves similar performance to masking diffusion, with the authors releasing their code for further exploration.

**Strengths:**

- The paper is well written, equations are nicely embedded and overall presentation is good (marking will be raised to good once the page limit is achieved).

- The paper effectively addresses limitations of existing discrete diffusion models, specifically the dominance of masking diffusion over more gradual and structured processes. It provides a solid theoretical foundation to argue for the introduction of SCUD.

- Introducing the SCUD model, which conditions on the transition schedule, is a novel approach. This model adapts diffusion frameworks by incorporating structured forward processes, potentially expanding discrete diffusion's applications across data types.

- The paper includes a rigorous theoretical framework, with proofs and mathematical propositions that justify the SCUD approach.

- The experiments span various data types, including images, text, and protein sequences. Results on CIFAR-10, LM1B, and UniRef50 datasets convincingly show that SCUD improves performance compared to other non-masking processes.

- The paper compares SCUD with state-of-the-art discrete diffusion models, showing how SCUD better captures transition schedules and can leverage structured processes.

**Weaknesses:**

# Presentation (Minor)

I will re-adjust my mark for presentation from poor to good once the following urgent concern has been addressed by the authors:

- The text of the main paper exceeds the 10-page limit. Please move your remarks regarding Reproducibility to the appendix, ICLR will likely be strict about enforcing the 10-page limit, exceeding the limit may lead to your work being rejected down the line.


# Content (Major)

- The paper's complexity could limit practical adoption. SCUD requires intricate parameterizations and careful handling of components like the rate parameter $\gamma$ and the transition matrix $K$ in particular, potentially increasing computational cost.

-  While SCUD reduces training samples, the discussion on costs associated with complex matrix operations, schedule conditioning, and increased dimensionality in high-dimensional data is minimal.

- The qualitative analysis of CIFAR-10 images suggests that SCUD models lack clear object formation. Though the focus is on likelihood scores, this limitation in sample quality could affect its utility in image generation tasks.

**Questions:**

In general, I am willing to raise my score, if my concerns and questions are addressed with compelling evidence.

Building on the aforementioned weaknesses, I pose the following questions:

1. Can you provide O-Notation w.r.t. the data-dimensionality for the added computational cost arising from $K$? Furthermore mentioning the GPU hours of the different methods in your work could help put the computational cost of different methods into perspective.

2. In addition to 1.: Could you provide a quantitative comparison of the computational costs associated with SCUD versus standard discrete diffusion methods in terms of memory usage and processing time?

3. Given the increased complexity SCUD introduces, what specific strategies could one use to manage computational demands when applying SCUD to larger datasets or higher-dimensional inputs?

4. Could you elaborate on how sensitive SCUD is to the selection of the rate parameter $\gamma $? How does this parameter interact with other hyperparameters in practice on datasets other than CIFAR-10 as shown in figure 2?

5. Are there particular domains where SCUD could have inherent advantages over masking diffusion?

6. Although SCUD improves likelihood scores, how do you plan to address the relatively low quality of visual samples generated in CIFAR-10? Could modifications to SCUD enhance sample fidelity?

---

> ### Author Response · Authors · 2024-11-24
> **Comment**
>
> Thank you for your thorough and thoughtful review! In our new draft, we incorporated your suggestions, 1) arguing and demonstrating that SCUD does not add significant computational overhead, 2) demonstrating in new experiments that SCUD can generate high quality images, and that SCUD is not sensitive to the choice of the rate parameter $\gamma$. We answer your questions while describing how we incorporate your suggestions below.
>
>
> >(W1) The text of the main paper exceeds the 10-page limit. Please move your remarks regarding Reproducibility to the appendix, ICLR will likely be strict about enforcing the 10-page limit, exceeding the limit may lead to your work being rejected down the line.
>
>
> We strongly appreciate your giving us a chance to fix such a potentially large formatting mistake! Our placement of the Reproducibility statement was informed by the following instructions from the [ICLR author’s guide](https://iclr.cc/Conferences/2025/AuthorGuide): “Authors are strongly encouraged to include a paragraph-long Reproducibility Statement at the end of the main text (before references) [sic] ... “ and “This optional reproducibility statement will not count toward the page limit, but should not be more than 1 page.”. Checking other papers, they seem to have the reproducibility statement in a similar place. Again thank you for bringing this to our attention!
>
> >(W2) The paper's complexity could limit practical adoption. SCUD requires intricate parameterizations and careful handling of components like the rate parameter  and the transition matrix  in particular, potentially increasing computational cost.
>
> >(W3) While SCUD reduces training samples, the discussion on costs associated with complex matrix operations, schedule conditioning, and increased dimensionality in high-dimensional data is minimal.
>
> >(Q1) Can you provide O-Notation w.r.t. the data-dimensionality for the added computational cost arising from ? Furthermore mentioning the GPU hours of the different methods in your work could help put the computational cost of different methods into perspective.
>
> >(Q2) In addition to 1.: Could you provide a quantitative comparison of the computational costs associated with SCUD versus standard discrete diffusion methods in terms of memory usage and processing time?
>
> To App. D.5 of the new draft we have added the below thorough discussion of considerations for 1) parameterization, 2) computational complexity from matrix computations and using $S$. We have included descriptions in big-O notation. We describe the strategies we used so that SCUD has minimal theoretical additional overhead for parameterization and computation compared to classical discrete diffusion, and in some cases allows new strategies for more efficient computation.
>
> Additionally, we’ve also added the GPU hours required to run SCUD and our implementations of masking and classical discrete diffusion to App. D.2,3,4 (all large models were trained for roughly 2 days each on 2 A100 GPUs on an academic cluster). In practice, we use virtually identical amounts of compute for our implementations of masking, SCUD, and classical discrete diffusion (maximum difference was classical diffusion was 20% faster than SCUD on images) and our GPU hours are comparable to previous methods with differences likely due to differences in hardware and implementation.
>
> 1. **Parameterization:**
>
> To specify a SCUD model, one needs to specify (1) the parameterization of the predictor $\tilde x_{0, \theta}$, (2) the rate function $\beta_t$, (3) a transition matrix $K$, (4) the amount to condition on the transition schedule $\gamma$. This is actually similar to what one must specify for classical discrete diffusion models: these need the parameterization and function (1) and (2); and, similar to (3), they must specify the infinitesimal generator $\mathcal L$. The major difference with SCUD is that $\gamma$ is set to $0$. Note below we show that SCUD is not very sensitive to the choice of $\gamma$, so there is little extra complexity involved in specifying a SCUD model compared to a classical discrete diffusion model.

---

> > ### Author Response · Authors · 2024-11-24
> > **Comment 2**
> >
> > 2: **Computational complexity:**
> >
> > In terms of computational complexity, the major differences are (A) replacing operations of $\mathcal L$ with operations of $K$, and (B) replacing the time $t$ in the argument of $\tilde x\_{0, \theta}$ with the number of transitions $S$. We discuss how (B) does not result in a large increase of computational complexity below, and note that (A) does not change the computational complexity except when the number of tokens $B$ is large, when it actually enables strategies that reduce complexity.
> >
> > 2A: *Matrix computations:*
> >
> > To calculate our loss, Eqn. 6, in Eqn. 7 we see that we only need to take matrix vector products with K; the analogous quantity in classical discrete diffusion requires matrix exponentiation $\exp(t\mathcal L)$ (Luo et al., 2022). When $B$ is small, both these calculations have negligible complexity and can be calculated similarly quickly by precomputing an eigen-decomposition of $K$ or $\mathcal L$. But when $B$ is large, as in the language modeling case, these calculations become very expensive; Luo et al., 2022 settled for very simple $\mathcal L$, masking and uniform, such that  $\exp(t\mathcal L)$ can be easily analytically calculated; SCUD is able to build in a richer forward process by picking a sparse + low rank $K$ so that matrix vector products are very fast.
> >
> > In terms of big-O notation, when an eigendecomposition is precomputed,  $(\exp(t\mathcal L) \tilde x\_0^d)\_{d=1}^D$ and $(K^{s\_t^d}\tilde x\_0^d)\_{d=1}^D$ each cost $\Theta(DB^2)$ for two dense matrix multiplies and a scaling by the exponentiation or power of the eigenvalues. When $K^{s\_t^d}$ is a sparse matrix with $O(rB)$ entries or has a rank of $r$, calculating $(K^{s\_t^d}\tilde x\_0^d)\_{d=1}^D$ is $O(DBr\max\_d(s\_t^d))$; in our language case, $B$ is large while $\max\_d(s\_t^d)\approx 30$ and we pick $r\approx 20$ resulting in a large speedup.
> >
> > 2B: *Computations with $S$:*
> >
> >  Indeed, the place that SCUD adds some overhead to calculations is in replacing the arguments of $\tilde x\_{0, \theta}(x\_t, \cdot)$: the time over which $x\_0$ has been corrupted, $t$, a scalar, is replaced with the number of corruptions of each token $S$, a $D$-dimensional object. The overhead of this operation is dependent on the architecture of $\tilde x\_{0, \theta}$. We picked $\tilde x\_{0, \theta}$ so that no parameters were added by replacing $t$ with $S$, and such that the computational and memory overhead caused by this replacement were negligible compared to the operations and memory spent on operations on the $D$-dimensional $x\_t$.
> >
> > In the paper we used previous architectures modified so that each operation on $t$ was also applied to each dimension of $S$. As well, for the architectures we chose, whenever a function of $t$ was added or multiplied to a set of activations, say at layer $\ell$, $h_{\ell, \theta}$, the activations had a dimension $D$, so we could perform the same operation with element-wise addition or multiplication with $S$, i.e.
> >
> > $$ h\_{\ell+1, \theta}^d= f\_{1, \theta}(t)h\_{\ell, \theta}^d+f\_{2, \theta}(t)$$
> >
> >  was replaced with
> >
> > $$ h\_{\ell+1, \theta}^d= f\_{1, \theta}(s\_t^d)h\_{\ell, \theta}^d+f\_{2, \theta}(s\_t^d).$$
> >
> > Thus, adapting $\tilde x\_{0, \theta}$ for SCUD in this way adds no extra parameters.
> >
> > The overhead of this change is that every call to $f\_{\theta}$ is replaced by $D$ calls, $D$-times the activations $f\_{\theta}(s\_t^d)$ must be stored, and $D$-times more gradients must be calculated for $f\_{\theta}(s\_t^d)$. $f\_{\theta}$ is however a set of linear functions and activations. The operations on the corrupted data $x\_t$ involve convolutions and attention, which have much larger memory and computational costs. In big-O notation, the cost of calculating $\tilde x\_{0, \theta}(x\_t, t)$ and $\tilde x\_{0, \theta}(x\_t, S)$ are therefore identical – at worst, the constant in front of the largest term changes. Therefore, in our experiments, we ran all models for roughly equal time with the same batch sizes and did not observe any substantial difference in computation.

---

> > > ### Author Response · Authors · 2024-11-24
> > > **Comment 3**
> > >
> > > >(Q3) Given the increased complexity SCUD introduces, what specific strategies could one use to manage computational demands when applying SCUD to larger datasets or higher-dimensional inputs?
> > >
> > > To recap the strategies mentioned above: we 1) use structured matrices $K$ for fast matrix-vector multiplies to accommodate large $B$ better than classical discrete diffusion, 2) treat $S$ as classical discrete diffusion treats $t$.
> > >
> > > While we argued above that (2) introduces minimal computational overhead, it is possible to reduce it further – note masking diffusion is an instance of SCUD and we could replicate its strategy to further improve computational efficiency. Masking takes $S$, identifies the dimensions with $s\_t^d>0$ and modifies those dimensions in $x\_t$ to get a masked $\hat x\_t$; it then makes it prediction $\tilde x_{0, \theta}(\hat x\_t)$. We can bake more information about $S$ into $x\_t$ as follows: the first step of $\tilde x\_{0, \theta}$ usually involves embedding each dimension of $x\_t$ into $\mathbb R^P$ for some $P$; we can similarly embed each dimension of $S$ and get a modified $\hat x\_t$ by adding or multiplying the representations of $x\_t^d$ and $s\_t^d$ and then predicting $\tilde x\_{0, \theta}(\hat x\_t)$ without the overhead of the high dimensional $S$. While more efficient, we anticipate however this is less flexible than adding $S$ into every layer as we did in the paper.
> > >
> > > >(Q5) Are there particular domains where SCUD could have inherent advantages over masking diffusion?
> > >
> > > We’ve demonstrated that SCUD, which generalizes masking, allows one to incorporate their inductive biases of the data into the forward process, while still incorporating known information about the transition schedule as in masking diffusion; in this paper we’ve used this to build models that outperform masking. In the future we anticipate that the flexibility of SCUD will allow practitioners to build models with more elaborate forward processes that model other types of data, such as insertion-deletion processes, and chemical graph data, while still conditioning on known information about the schedule.
> > >
> > > >(W4)The qualitative analysis of CIFAR-10 images suggests that SCUD models lack clear object formation. Though the focus is on likelihood scores, this limitation in sample quality could affect its utility in image generation tasks.
> > >
> > > >(Q6) Although SCUD improves likelihood scores, how do you plan to address the relatively low quality of visual samples generated in CIFAR-10? Could modifications to SCUD enhance sample fidelity?
> > >
> > > In Fig. 4 we showed samples from SCUD-Gaussian. While these samples were more realistic than autoregressive models, they did not form recognizable objects like samples from diffusion models with worse perplexities. In [Fig S1](https://anonymous.4open.science/api/repo/host_image-6044/file/comparison.pdf?v=ebfdc962) of the global response, we show that when SCUD is trained for longer, it generates much higher quality images that often contain recognizable animals or vehicles. While there are a number of straightforward methods that may improve sample quality, such as informed or uninformed corrector steps (Zhao et al, 2024, Campbell et al, 2022) or swapping the infinitesimal generator at test time (Campbell et al., 2024, explained in our new App. E) training for longer may be sufficient to achieve good image quality as well as perplexity.

---

> > > > ### Author Response · Authors · 2024-11-24
> > > > **Comment 4**
> > > >
> > > > >(Q4) Could you elaborate on how sensitive SCUD is to the selection of the rate parameter ? How does this parameter interact with other hyperparameters in practice on datasets other than CIFAR-10 as shown in figure 2?
> > > >
> > > > In Sec. 5 we show that $\gamma$ continuously interpolates between full schedule conditioning at $\gamma=1$ and classical discrete diffusion at $\gamma=0$. Therefore we expect performance of our model to interpolate between our $\gamma=1$ models and the performance of classical discrete diffusion as $\gamma$ goes from $1$ to $0$. Empirically, this is what we see in Fig. 2 for two different forward processes; we further see that the interpolation is roughly monotonic.
> > > >
> > > > Other experiments in Sec. 7 show that the choice of $\gamma=1$ improves upon the classical choice of $\gamma=0$. Therefore a practitioner may safely pick $\gamma=1$ without tuning.
> > > >
> > > >  In a new experiment we check if there is a dramatically better choice of $\gamma$ than $0$ or $1$ in the protein case. We trained 11 models with $\gamma=0, 0.1, \dots, 0.9, 1.0$ for 3 epochs of Uniref50 (1 GPU for 14 h each); note that the results are noisy as we did not have the resources to run replicates. In this small scale setting, the $\gamma=0.0$ and $\gamma=1.0$ models both had the same negative log likelihood to the third digit (2.768). We now asked: if a practitioner attempted to tune $\gamma$ between $0$ and $1$, would they get a much better result? This experiment suggests the answer is no, as the best run at $\gamma=0.3$ has a nearly identical negative log likelihood of 2.765 (the worst was 2.777 ($\gamma=0.8)). Thus for both CIFAR and protein, our results suggest that $\gamma=1$ is a safe choice for the practitioner.
> > > >
> > > > **Citations:**
> > > >
> > > > Zhao, Yixiu, Jiaxin Shi, Lester Mackey, and Scott Linderman. 2024. “Informed Correctors for Discrete Diffusion Models.” arXiv [Cs.LG]. arXiv. http://arxiv.org/abs/2407.21243.
> > > >
> > > > Campbell, Andrew, Joe Benton, Valentin De Bortoli, Tom Rainforth, George Deligiannidis, and Arnaud Doucet. 2022. “A Continuous Time Framework for Discrete Denoising Models.” In Advances in Neural Information Processing Systems. https://openreview.net/pdf?id=DmT862YAieY.
> > > >
> > > > Campbell, Andrew, Jason Yim, Regina Barzilay, Tom Rainforth, and Tommi Jaakkola. 2024. “Generative Flows on Discrete State-Spaces: Enabling Multimodal Flows with Applications to Protein Co-Design.” In Proceedings of the 41st International Conference on Machine Learning. arXiv. https://doi.org/10.48550/ARXIV.2402.04997.

---

> > > > > ### Comment · Reviewer_bGrB · 2024-11-25
> > > > >
> > > > > I want to thank the authors for their detailed responses.
> > > > >
> > > > > My concerns regarding computational complexity have been fully addressed, however I remain unconvinced regarding (Q5) & (Q6). Figure S1 does not yield convincing visual assessments. The authors claim that their method outperforms masking, however, SCUD is outperformed by other methods in all Tables (1-3).
> > > > >
> > > > > Therefore, I maintain my initial assessment (6: marginally above the acceptance threshold).

---

> > > > > > ### Author Response · Authors · 2024-12-02
> > > > > > **Response**
> > > > > >
> > > > > > Thank you for your response! We believe we can give a more satisfying answer of (Q5) and (Q6) with new experiments in [the global response](https://openreview.net/forum?id=wQk6yaRGOi&noteId=7Bx81CwgC8).
> > > > > >
> > > > > > >(Q6) Although SCUD improves likelihood scores, how do you plan to address the relatively low quality of visual samples generated in CIFAR-10? Could modifications to SCUD enhance sample fidelity?
> > > > > >
> > > > > > Samples from fully trained D3PM and $\tau$LDR models resemble the objects in CIFAR10. The last set of examples from SCUD, although much better than the first iteration, did not look as good as samples from these previous models ([FigS1](https://anonymous.4open.science/api/repo/host_image-6044/file/comparison.pdf?v=ebfdc962)). Our new samples, in [Fig S2](https://anonymous.4open.science/api/repo/host_image-6044/file/SCUD%20samples%20Dec%201.png?v=1f77a1db) much more clearly show vehicles and animals.
> > > > > >
> > > > > > >(Q5) Are there particular domains where SCUD could have inherent advantages over masking diffusion?
> > > > > >
> > > > > > Our hypothesis was that those methods in the tables that outperform SCUD only do so because they are trained on much more data. We did not have the computational resources to train SCUD for the same amount of time as previous methods have – some of these methods are trained on weeks or months of GPU time. However, to support our hypothesis, we demonstrated SCUD outperforms these previous methods (masking and classical discrete diffusion) when trained on the same amount of data.
> > > > > >
> > > > > > Nevertheless, during this rebuttal period we have trained SCUD for much longer; in our [updated tables](https://openreview.net/forum?id=wQk6yaRGOi&noteId=7Bx81CwgC8) we now show that when trained for sufficiently long, SCUD indeed outperforms the previous methods on proteins and images.

---

### Official Review · Reviewer_QQ5r · 2024-11-06

**Soundness:** 3
**Presentation:** 3
**Contribution:** 3
**Rating:** 8
**Confidence:** 4

**Summary:**

The choice of forward process has great performance implications for (discrete) diffusion models. This paper theoretically studies the improved performance of the masking forward process in discrete diffusion models. To this end, the authors introduce the notion of an event schedule, which describe times along the forward process where transitions take place, and separate the "when" of transition from the "where" of transition.

The authors further derive the training objectives corresponding to conditioning on these event schedules, and apply their proposed method SCUD, to the tasks of image generation, language generation and protein sequence generation. Across the different tasks, their proposed method is able to outperform equivalent forward processes, but without the conditioning using less training examples.

**Strengths:**

1. The main strength of the paper is the firm theoretical footing to understand different forward processes in discrete diffusion, and how they relate to the "when" and "where" of the corresponding transitions. Through the notion of event / schedule conditioning, the paper attempts to disentangle the influence of "when" and "where", which leads to corresponding modifications to the ELBO objective. The authors also emphasize the connections of their method to previous discrete diffusion methods.

2. The paper is very well written, and makes a strong effort to coherently explain the different moving parts.

3. To connect the method to practice, the authors also propose different tricks such as reversing multiple events jointly, and an efficient loss formulation for high-dimensional data. Experiments across image, protein and language domain show favorable improvements for the same forward process, conditioned on event schedule. The experiments on proteins are especially interesting, using 2 orders less of training data.

**Weaknesses:**

There are no glaring weaknesses in the paper. But there is some minor feedback:

1. The formal notion of event schedule is only introduced in Proposition 4.2. This should instead be moved to before Proposition 3.1, so the readers already know what the event schedule captures, and the corresponding equations become easier to follow.

2. There are claims in the paper regarding SCUD outperforming masking, but evidence of this is not visible in the experiments. It is unclear whether the SCUD conditioning with Gaussian / Uniform will outperforming existing discrete diffusion (e.g. D3PM) with masking, given equal compute.

**Questions:**

1. In light of the point 2 in weaknesses, could the authors run the D3PM model with masking diffusion with the same number of training samples as SCUD? -- The reviewer hopes this is feasible within the available compute budget of the authors. An alternative would be to also consider small versions of D3PM and SCUD that can be trained for equivalent number of samples.

2. How is B defined in the language and protein experiments?

---

> ### Author Response · Authors · 2024-11-24
> **Comment**
>
> Thank you for your thorough and thoughtful review! We have incorporated your writing and experimental advice into the new draft. We respond to your individual suggestions below.
>
> >(W1) The formal notion of event schedule is only introduced in Proposition 4.2. This should instead be moved to before Proposition 3.1, so the readers already know what the event schedule captures, and the corresponding equations become easier to follow.
>
> We have moved this definition to the sentence before Prop. 3.1.
>
> >(W2) There are claims in the paper regarding SCUD outperforming masking, but evidence of this is not visible in the experiments. It is unclear whether the SCUD conditioning with Gaussian / Uniform will outperforming existing discrete diffusion (e.g. D3PM) with masking, given equal compute.
>
> >(Q1) In light of the point 2 in weaknesses, could the authors run the D3PM model with masking diffusion with the same number of training samples as SCUD? -- The reviewer hopes this is feasible within the available compute budget of the authors. An alternative would be to also consider small versions of D3PM and SCUD that can be trained for equivalent number of samples.
>
> In our paper we claim that SCUD generalized masking diffusion and this generalization allows it to incorporate inductive biases in the forward process that improve model performance. We showed mathematically that SCUD with a uniform forward process with $\gamma=1-1/B$ is masking. We showed explicitly in Fig. 2 that SCUD outperforms masking. In the rest of the experiments, we compared SCUD with structured forward processes with SCUD with a uniform forward process; this later model we argued was similar to masking, which we see evidence of in Fig. 2.
>
> To properly justify our claims comparing SCUD and masking diffusion, in new results on images and proteins in the [global response](https://openreview.net/forum?id=wQk6yaRGOi&noteId=4fknjBKsjZ), we run our results for longer and replace the SCUD uniform baseline with explicit masking diffusion as in Fig. 2. We indeed see that SCUD with structured forward processes outperforms our reimplemented masking diffusion.
>
> >(Q2) How is B defined in the language and protein experiments?
>
> For the protein experiments, $B$ is defined as the number of amino acids $20$ plus $11$ special tokens that appear in the data representing ambiguous amino acids. For language, $B$ is the number of text tokens, which is 30522 for the BERT tokenizer. We have added this into the description of experiments in Sec. 7.

---

> > ### Comment · Reviewer_QQ5r · 2024-11-25
> > **Thank you for your response**
> >
> > I thank the authors for the their response and clarifications.
> >
> > > (W2) There are claims in the paper regarding SCUD outperforming masking, but evidence of this is not visible in the experiments. It is unclear whether the SCUD conditioning with Gaussian / Uniform will outperforming existing discrete diffusion (e.g. D3PM) with masking, given equal compute.
> >
> > Just for clarification, this comment was made by looking at the experiment section, rather than Figure 2. The experiments I was interested in is seeing were:
> >
> > * For proteins, D3PM with Masking forward process for the same number of training tokens as SCUD
> > * For language SEDD with Masking forward process and same number of training tokens as SCUD.
> >
> > I assume the Classical with Masking forward process experiment now added is equivalent to D3PM, and thus my question would be addressed.
> >
> > (Minor): It would have been ideal to use a different color like blue to highlight the revisions.
> >
> > My initial assessment of the paper was already on the positive side, and I will be maintaining the score after the rebuttal.

---

> > > ### Author Response · Authors · 2024-12-02
> > > **Response**
> > >
> > > Thank you for your response!
> > >
> > > For proteins (as well as images), we felt it would be most fair to compare with an implementation of masking that represents a model that a knowledgeable practitioner might train as an alternative to SCUD. This model has nearly the same implementation as SCUD but uses masking ($\gamma=1-1/B$ and an averaged loss in Prop. 5.1) instead of the structured forward process. It is similar to more recent continuous time discrete diffusion models – SEDD, MD4, MDLM – which have been shown to dramatically outperform the early implementation of D3PM; for this reason, we felt it would be unfair to compare directly to a reimplementation of D3PM.
> > >
> > > Nevertheless, you are asking for evidence that if we trained for longer we would beat the numbers from previous papers. After running our image and protein models for longer in the [global response](https://openreview.net/forum?id=wQk6yaRGOi&noteId=7Bx81CwgC8), we indeed see that SCUD outperforms the previous baselines on images and proteins.

---

### Author Response · Authors · 2024-11-24
**Global comment**

We thank all reviewers for their thoughtful reviews! There is a flurry of work extending the capacities and design space of discrete diffusion models. However, in many settings, the simplest diffusion model, masking diffusion, dominates. Therefore, designs and capacities deviating from masking diffusion, for example, incorporate inductive biases in the forward process, often harm rather than help inference. SCUD explains the surprising dominance of masking diffusion. We use this insight to incorporate inductive biases into the forward process without sacrificing schedule conditioning, thus building better models. SCUD opens the door to moving beyond masking diffusion and leveraging the full design space of discrete diffusion.

Below we discuss how we have incorporated each of your suggestions into the new draft of the paper, adding a discussion on the computational complexity of SCUD, and extending SCUD to flow matching. In addition to these changes we have run our image (B=256) and protein experiments for longer along with our own careful implementations of masking diffusion and classical diffusion baselines. These reimplementations use the same training and architecture as SCUD and have strong performance on their own. Nevertheless, the results (Tab 1, 2) show that SCUD outperforms these baselines and the longer training improved the performance of SCUD. [FigS1](https://anonymous.4open.science/api/repo/host_image-6044/file/comparison.pdf?v=ebfdc962) also shows that the sample quality of SCUD greatly improves with longer training. These models will continue to run until the end of the discussion section, when we will update these numbers.

**Table S1: Discrete diffusion models on image data**
|Method|Forward process|Training samples|BPD|
|-|-|-|-|
|D3PM|Uniform|1.9×10⁸|5.08|
|D3PM|Gaussian|1.9×10⁸|3.44|
|τLDR|Gaussian|2.6×10⁸|3.59|
|MD4|Masking|2.6×10⁸|**2.78**|
|Classical|Gaussian|6.4×10⁷|2.94|
|Classical|Masking|6.4×10⁷|2.90|
|SCUD|Gaussian|6.4×10⁷|*2.86*|


**Table S2: Discrete diffusion models on protein data**
|Method|Forward process|Training tokens|Perplexity|
|-|-|-|-|
|D3PM|Uniform|>3×10¹¹|18.82|
|D3PM|BLOSUM|>3×10¹¹|17.16|
|D3PM|Masking|>3×10¹¹|**14.61**|
|Classical|BLOSUM|8×10⁹|15.39|
|Classical|Masking|8×10⁹|15.56|
|SCUD|BLOSUM|8×10⁹|*15.29*|

---

> ### Author Response · Authors · 2024-12-02
> **Global comment on updated numbers**
>
> We thank all reviewers for a productive review session! SCUD makes an important theoretical contribution to discrete diffusion models by explaining the dominance of masking diffusion. In addition, we use this insight to build models that outperform masking in practice. In experiments we showed that by combining schedule conditioning and a structured forward process, SCUD outperforms 1) our careful implementations of classical discrete diffusion and masking diffusion, and 2) previously reported numbers for classical discrete diffusion methods. Previously reported numbers for masking diffusion came from models that were trained for weeks or months of GPU time which we could not achieve.
>
> Reviewers suggested that it would strengthen the paper to beat these numbers as well. Therefore, throughout this review session we have run our methods for much longer. Tab 1, 2 show that when trained for sufficiently long SCUD indeed outperforms all previously trained models as we hypothesized. [FigS2](https://anonymous.4open.science/api/repo/host_image-6044/file/SCUD%20samples%20Dec%201.png?v=1f77a1db) also shows that the sample quality of SCUD again greatly improves with longer training, now containing recognizable vehicles and animals. With this new set of results, SCUD achieves state-of-the-art likelihoods among discrete diffusion models on two modalities.
>
> Table 1
> |Method|Forward process|Training samples|BPD|
> |-|-|-|-|
> |D3PM|Uniform|1.9×10⁸|5.08|
> |D3PM|Gaussian|1.9×10⁸|3.44|
> |τLDR|Gaussian|2.6×10⁸|3.59|
> |MD4|Masking|2.6×10⁸|2.78|
> |Classical|Gaussian|1.3×10⁸|2.83|
> |Classical|Masking|1.3×10⁸|2.81|
> |SCUD|Gaussian|1.3×10⁸|**2.77**|
>
> Table 2
> |Method|Forward process|Training tokens|Perplexity|
> |-|-|-|-|
> |D3PM|Uniform|>3×10¹¹|18.82|
> |D3PM|BLOSUM|>3×10¹¹|17.16|
> |D3PM|Masking|>3×10¹¹|14.61|
> |Classical|BLOSUM|1×10¹¹|14.70|
> |Classical|Masking|1×10¹¹|14.86|
> |SCUD|BLOSUM|1×10¹¹|**14.54**|

---

### Meta-Review · Area_Chair_kJUg · 2024-12-20

**Metareview:**

The current paper proposes to condition on the transition schedule in the forward-pass of discrete diffusion models. The reviewers acknowledged that the authors' work gives novel theoretical insights into the field and that the overall presentation is coherent. The reviewers' questions regarding the computational cost have been addressed during the rebuttal. In the final discussion, there are still major concerns on the experimental evidence including visual qualities of the generated images and a thorough comparison with masking diffusion under a fair evaluation protocol, and thus the current empirical evaluation is not conclusive. Moreover, Reviewer bGrb suggested to study how fast SCUD converges relative to other methods based on checkpoints of the models over time, which is not included in the current paper. To incorporate these additional results would need more efforts and further validation, and we cannot be certain whether this can be accomplished within the tight timeline of ICLR'25 but hope will be achieved in the next iteration of this work. The rejection is based on careful evaluation and thorough discussion with regret. The authors are encouraged to resubmit the paper after the suggested revisions with improved and more stable empirical results.

**Additional Comments On Reviewer Discussion:**

**UPDATE Jan 9 2025**: the meta review has been extended to indicate the paper has been handled thoroughly.

The review by reviewer 3fup is considered less important due to low confidence as requested by the reviewer themselves.

In the reviewer discussion phase, two reviewers bGrB and ifSU raised major concerns on their experimental results (although giving a score of 6). The other reviewer QQ5r (giving a score of 8) did not comment in this discussion, but did mention that the experimental evidence is not visible in their original review.

The insufficient empirical evaluation seems to be due to lack of computational resources as the authors have mentioned in the rebuttal. This work can benefit from another round of revision with more solid empirical validation.

---

### Decision · Program_Chairs · 2025-01-22

Reject